# Proximity proteomics in a marine diatom reveals a putative cell surface-to-chloroplast iron trafficking pathway

Jernej Turnšek[1,2,3,4,5,6†], John K Brunson[6,7], Maria del Pilar Martinez Viedma[8‡], Thomas J Deerinck[9], Aleš Horák[10,11], Miroslav Oborník[10,11], Vincent A Bielinski[12], Andrew Ellis Allen[4,6]*

[1]Biological and Biomedical Sciences, The Graduate School of Arts and Sciences, Harvard University, Cambridge, United States; [2]Department of Systems Biology, Harvard Medical School, Boston, United States; [3]Wyss Institute for Biologically Inspired Engineering, Harvard University, Boston, United States; [4]Integrative Oceanography Division, Scripps Institution of Oceanography, University of California San Diego, La Jolla, United States; [5]Center for Research in Biological Systems, University of California San Diego, La Jolla, United States; [6]Microbial and Environmental Genomics, J. Craig Venter Institute, La Jolla, United States; [7]Center for Marine Biotechnology and Biomedicine, Scripps Institution of Oceanography, University of California San Diego, La Jolla, United States; [8]Informatics, J. Craig Venter Institute, La Jolla, United States; [9]National Center for Microscopy and Imaging Research, University of California San Diego, La Jolla, United States; [10]Biology Centre CAS, Institute of Parasitology, České Budějovice, Czech Republic; [11]University of South Bohemia, Faculty of Science, České Budějovice, Czech Republic; [12]Synthetic Biology and Bioenergy, J. Craig Venter Institute, La Jolla, United States

*For correspondence:
aallen@ucsd.edu

Present address: †Department of Plant and Microbial Biology, University of California, Berkeley, Howard Hughes Medical Institute, University of California, Berkeley, United States; ‡PharmaMar S.A., Madrid, Spain

Competing interests: The authors declare that no competing interests exist.

**Abstract** Iron is a biochemically critical metal cofactor in enzymes involved in photosynthesis, cellular respiration, nitrate assimilation, nitrogen fixation, and reactive oxygen species defense. Marine microeukaryotes have evolved a phytotransferrin-based iron uptake system to cope with iron scarcity, a major factor limiting primary productivity in the global ocean. Diatom phytotransferrin is endocytosed; however, proteins downstream of this environmentally ubiquitous iron receptor are unknown. We applied engineered ascorbate peroxidase APEX2-based subcellular proteomics to catalog proximal proteins of phytotransferrin in the model marine diatom *Phaeodactylum tricornutum*. Proteins encoded by poorly characterized iron-sensitive genes were identified including three that are expressed from a chromosomal gene cluster. Two of them showed unambiguous colocalization with phytotransferrin adjacent to the chloroplast. Further phylogenetic, domain, and biochemical analyses suggest their involvement in intracellular iron processing. Proximity proteomics holds enormous potential to glean new insights into iron acquisition pathways and beyond in these evolutionarily, ecologically, and biotechnologically important microalgae.

## Introduction

Iron (Fe) likely played an important role in the origin of life (*Bonfio et al., 2017*; *Jin et al., 2018*) and is fundamental to almost all extant metabolisms (*Aguirre et al., 2013*) serving as a cofactor in enzymes involved in DNA replication, photosynthesis, cellular respiration, nitrate assimilation,

nitrogen fixation, and reactive oxygen species defense (*Crichton, 2016*). Early anoxic oceans were rich in readily bioavailable ferrous, $Fe^{2+}$, iron, but with the rise of oxygenic photosynthesis and the Great Oxygenation Event (GOE) in the Paleoproterozoic ~2.3 billion years ago followed by the Neo-proterozoic Oxygenation Event (NOE) ~1.5 billion years later, most of the ferrous iron oxidized into insoluble ferric, $Fe^{3+}$, minerals which are not bioavailable (*Camacho et al., 2017*; *Knoll et al., 2016*; *Och and Shields-Zhou, 2012*). Conceivably, these large global shifts in ocean chemistry could have had a major role in driving the evolution of novel iron uptake mechanisms.

Iron availability limits primary productivity in high-nutrient, low-chlorophyll (HNLC) regions which cover ~25% of the modern ocean habitat as demonstrated by numerous large-scale Fe fertilization experiments and natural Fe upwelling events invariably resulting in diatom-dominated phytoplankton blooms (*Ardyna et al., 2019*; *de Baar, 2005*; *Martin et al., 1994*). Although dissolved Fe in marine environments is primarily found complexed to organic ligands (*Hutchins and Boyd, 2016*; *Tagliabue et al., 2017*) such as bacterially produced siderophores and hemes (*Boiteau et al., 2016*; *Hogle et al., 2014*), low (pM) amounts of unchelated labile Fe, Fe', are crucial for eukaryotic phytoplankton, particularly diatoms (*Morel et al., 2008*), widespread single-celled microalgae critical to oceanic primary production (*Benoiston et al., 2017*). Diatoms have convergently evolved phyto-transferrin (pTF), which serves as the basis of a high-affinity ferric, $Fe^{3+}$, iron binding, and acquisition pathway (*McQuaid et al., 2018*; *Morrissey et al., 2015*). Phytotransferrin sequences have a broad taxonomic distribution and are abundant in marine transcriptomic datasets (*Bertrand et al., 2015*; *Marchetti et al., 2012*).

Phytotransferrins are estimated to have emerged concurrently with the NOE exemplifying the link between large environmental changes and molecular innovation (*McQuaid et al., 2018*). They are phylogenetically related to Fe-assimilation (FEA) domain-containing proteins found in freshwater and marine single-celled algae, and represent functional analogues of iron delivery proteins transferrins (*Behnke and LaRoche, 2020*; *Cheng et al., 2004*; *McQuaid et al., 2018*; *Scheiber et al., 2019*). pTF (Ensembl ID: Phatr3_J54465, UniProt ID: B7FYL2) from the model marine diatom *Phaeodactylum tricornutum* (*Bowler et al., 2008*) was first identified as an iron deficiency-sensitive gene and named *ISIP2a* (iron starvation induced protein 2a) (*Allen et al., 2008*), although its expression is relatively high in iron-replete conditions as well (*Smith et al., 2016*). pTF localizes to the cell surface and intracellular puncta, presumably endosomal vesicles (*McQuaid et al., 2018*), which is further supported by the presence of an endocytosis motif in the C-terminus of the protein (*Lommer et al., 2012*). In diatom pTF, dissolved labile ferric iron is coordinated synergistically with carbonate anion ($CO_3^{2-}$), a common feature of all transferrins (*Cheng et al., 2004*; *McQuaid et al., 2018*; *Zak et al., 2002*). Thus, decline in seawater carbonate concentrations due to ocean acidification caused by elevated atmospheric $CO_2$ may negatively impact this prevalent iron uptake system in diatoms and other marine phytoplankton (*McQuaid et al., 2018*).

While accessing and binding dilute dissolved labile ferric iron on the cell surface is important, its subsequent internalization and delivery to target sites within complex cellular milieu are also critical. $Fe^{3+}$ is highly insoluble and conversion between $Fe^{3+}$ and $Fe^{2+}$ can lead to toxic reactive oxygen species causing damage to proteins, lipids, and nucleic acids (*Cheng et al., 2004*). Precise control of iron internalization and intracellular trafficking in either of its redox states is thus crucial for maintenance of cellular homeostasis (*Philpott and Jadhav, 2019*; *Wang and Pantopoulos, 2011*). Human ferric iron-laden transferrin (Tf) bound to transferrin receptor is internalized via clathrin-mediated endocytosis. Endosome acidification leads to carbonate protonation and Tf conformational change resulting in iron release. $Fe^{3+}$ is then reduced by six-transmembrane epithelial antigen of prostate 3 (STEAP3), exported to the cytoplasm through divalent metal transporter 1 (DMT1; also NRAMP2) or Zrt- and Irt-like protein (ZIP) 14, and offloaded to iron chaperones for distribution to cellular iron sinks (*Bogdan et al., 2016*; *Eckenroth et al., 2011*; *Ohgami et al., 2005*; *Philpott and Jadhav, 2019*; *Wang and Pantopoulos, 2011*). In contrast, proteins that mediate intracellular allocation of internalized $Fe^{3+}$ and conduct specific biochemical transformations downstream of diatom pTF are unknown. Elucidation of these pathways will advance our understanding of diatom adaptation to survival in iron-deficient ocean waters and provide insight into how climate change will impact diatom physiology and fitness.

To investigate the proximal proteomic neighborhood of *P. tricornutum* pTF, we employed APEX2, an engineered soybean ascorbate peroxidase that functions as a dual probe for electron microscopy and proximity proteomics (*Hung et al., 2016*; *Lam et al., 2015*; *Martell et al., 2017*).

When fused with a protein of interest (bait protein), the ~27 kDa APEX2 enzyme permits spatially resolved proteomic mapping by oxidizing biotin-phenol to short lived (<1 ms) phenoxyl radicals which can covalently react with electron-rich amino acid residues—primarily tyrosine—on the surface of nearby endogenous proteins. Tagged proteins can then be isolated by purification with streptavidin beads and analyzed using mass spectrometry (MS). While the 'biotinylation radius' in APEX2 experiments is estimated to be in the 10–20 nm range, it should rather be seen as a 'probability gradient' where the likelihood of tagging decreases with distance away from the APEX2-tagged protein of interest (*Gingras et al., 2019*; *Hung et al., 2016*; *Lam et al., 2015*). Proteomic hits can therefore include vicinal proteins which interact with the APEX2-tagged bait protein indirectly in addition to direct interactors of varying strength (*Lundberg and Borner, 2019*). Tha ability to capture weakly and transiently interacting proteins in a biological process of interest is a major advantage APEX2 offers over traditional pull-down approaches (*Gingras et al., 2019*). Swaping biotin-phenol for diaminobendizine (DAB)—another APEX2 substrate—enables one to determine fusion protein localization via high-resolution electron microscopy making APEX2 a powerful bifunctional probe (*Martell et al., 2017*; *Martell et al., 2012*).

APEX2 and related enzyme-based chemical biology tools have over the past decade been applied to investigate cellular compartments and processes in diverse model systems including mammalian cell culture, yeast, protists, and plant protoplasts (*Gingras et al., 2019*; *Trinkle-Mulcahy, 2019*). APEX2 was recently used to map proteomes associated with post-Golgi vesicle trafficking and endocytosis (*Del Olmo et al., 2019*; *Otsuka et al., 2019*) as well as receptor-mediated signaling (*Paek et al., 2017*), successes that are all directly pertinent to the goals of our study.

Here, we provide further evidence for *P. tricornutum* phytotransferrin pTF endocytosis, demonstrate correct subcellular localization and activity of its APEX2 fusion, and identify nearly 40 likely proximal proteins through a proximity-dependent proteomic mapping experiment conducted in quintuplicate. To the best of our knowledge, this study represents the first application of APEX2 in a marine microbial model system (*Figure 1*). We then provide initial characterization, using dual fluorophore protein tagging fluorescence microscopy, phylogenetic, domain, and biochemical analyses, for three proteins expressed from a chromosomal gene cluster (pTF.CREG1, pTF.CatCh1, pTF.ap1) believed to act in association with endocytosed pTF downstream of ferric iron binding at the cell surface. Bacterially expressed recombinant pTF.CREG1 was found to be catalytically dead indicating it may have a regulatory, rather than enzymatic, role in pTF endocytosis, similar to functions proposed for human CREG1. Meanwhile, bioinformatic interrogation of pTF.CatCh1 and its localization pattern suggest it might be a chloroplast-associated iron-binding protein. Finally, we introduce a model for an intracellular ferric iron allocation pathway in *P. tricornutum* based on our proximity proteomics hits.

## Results

### *P. tricornutum* phytotransferrin (pTF) is localized to intracellular vesicles

To confirm pTF association with intracellular vesicles as suggested by *McQuaid et al., 2018*, a pTF-mCherry encoding episome was conjugated into Δ*pTF P. tricornutum* cells (*Diner et al., 2016*; *Karas et al., 2015*; *McQuaid et al., 2018*). After labeling the fluorescent transconjugant cell line with 100 µM of the membrane dye MDY-64 for 10 min at room temperature, colocalized mCherry and MDY-64 signals were observed on the cell surface and within intracellular vesicles (*Figure 2A*).

### APEX2 fused with *P. tricornutum* phytotransferrin (pTF) is enzymatically active in vivo

To generate pTF-APEX2 expressing diatom cell lines, an episome encoding pTF with C-terminal APEX2 was assembled and conjugated into WT *P. tricornutum* cells (*Figure 2B*). Considering the predicted pTF domains (*Figure 2—figure supplement 1A*), APEX2 was likely facing the cytosol at the cell surface and once internalized into vesicles. Western blot with pTF-specific antibodies (*McQuaid et al., 2018*) confirmed the ~84.7 kDa fusion protein expression in four out of five tested transconjugant cell lines. The protein was present in the insoluble cell lysate fraction further indicative of its membrane localization (*Figure 2C*). Amplex UltraRed, a highly sensitive APEX2 substrate

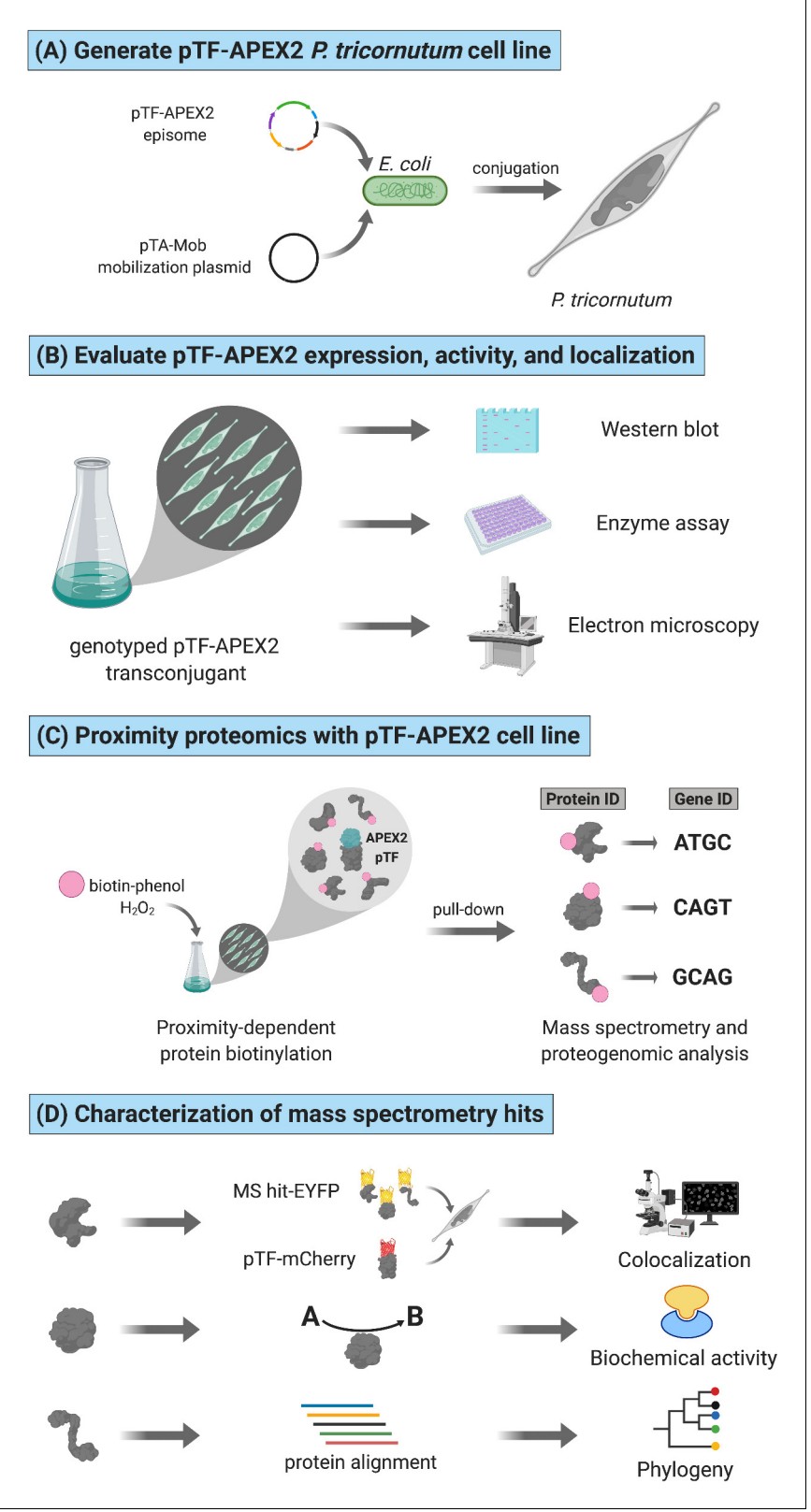

**Figure 1.** Functional APEX2-enabled proteogenomics with pTF (phytotransferrin/ISIP2a/Phatr3_J54465) in the model marine diatom *Phaeodactylum tricornutum*. (**A**) pTF-APEX2 encoding episome is introduced into *P. tricornutum* cells using bacterial conjugation. (**B**) Resulting transconjugants are genotyped and evaluated for fusion protein expression. APEX2 activity and fusion protein localization are then confirmed with an enzymatic assay and
*Figure 1 continued on next page*

*Figure 1 continued*

electron microscopy, respectively. (C) In a proximity-dependent proteomic mapping (proximity proteomics) experiment, pTF-APEX2 expressing cell line is supplemented with biotin-phenol and hydrogen peroxide, reaction quenched, and cells lysed. Cell lysate is then subjected to streptavidin pull-down, proteins analyzed with mass spectrometry (MS), peptides mapped to a *P. tricornutum* proteome database, and corresponding genes identified. (D) Interesting MS hits are further evaluated experimentally (e.g. for colocalization with the bait protein (i.e. pTF) and/or for predicted biochemical activity) as well as bioinformatically. Created with BioRender.com.

(*Hung et al., 2016*), was used to assay live pTF-APEX2 expressing cells. A resorufin (reaction product of Amplex UltraRed and APEX2) signal up to 4- and 40-fold above WT background was observed in experiments performed at room temperature (data not shown) and on ice (*Figure 2D*; *Figure 2— source data 1*), respectively, indicating active APEX2 with incorporated heme. Resorufin was also directly visualized by confocal microscopy and a strong cytosolic signal not tightly localized to the expected site of origin, similar to previous reports (*Martell et al., 2012*), was observed (*Figure 2— figure supplement 1B*), perhaps indirectly supporting our predicted APEX2 orientation. APEX2 was active only in the presence of both Amplex UltraRed and hydrogen peroxide (*Figure 2—figure supplement 1C*) implying that endogenous $H_2O_2$ levels are not sufficient to drive APEX2-catalyzed reactions, and that the overall cell surface and intracellular milieu in *P. tricornutum* is permissive to the APEX2 catalytic cycle.

## pTF-APEX2 is localized to the cell membrane and intracellular vesicles

To confirm pTF-APEX2 is localized to the cell surface and intracellular vesicles similar to the pTF-mCherry fusion protein, pTF-APEX2 expressing cells (cell line s2, *Figure 2C–D*) were treated with 25 mM 3,3'-diaminobenzidine (DAB) in the presence of 3 mM $H_2O_2$ for 15 min on ice (*Figure 3A*). This reaction leads to DAB polymerization and local precipitation around APEX2 that can be stained with osmium tetroxide and visualized with an electron microscope. Cells were embedded in a 3% agar matrix to prevent losses during numerous washing steps (*Figure 3—figure supplement 1A*). Tightly localized signal was observed on the cell membrane and in intracellular vesicles indicating that pTF-APEX2 fusion trafficked to the correct subcellular sites (*Figure 3B*). Additionally, we observed mitochondrial signal in WT and pTF-APEX2 cell lines subjected to the DAB reaction (*Figure 3B*; *Figure 3—figure supplement 1B*). Analysis of the *P. tricornutum* proteome revealed eight peroxidases with Arg38, His42, His163, and Asp208; catalytic amino acid residues that are conserved across all identified ascorbate peroxidases (APX) including soybean APX (*Raven, 2003*) and its derivative APEX2 (*Figure 3—figure supplement 1C*). Three of them contain proline in place of alanine at position 134 relative to APEX2, a substrate-binding loop mutation rendering APEX2 highly active (*Lam et al., 2015*), and another one is predicted to localize to mitochondria (*Figure 3—figure supplement 1D–E*). It is plausible that one or more of these endogenous APEX2-like peroxidases are responsible for the observed mitochondrial signal. APEX2 was inactivated in iron-deplete conditions (40 nM total Fe) and could be reactivated by supplementing fixed cells with 10 µM hemin chloride for 3 hr (data not shown), but this substantially increased background in the Amplex UltraRed assay. Therefore, subsequent proximity labeling experiments were carried out on cells growing in iron-replete conditions.

## Identification of the proximal phytotransferrin proteome with biotin-phenol labeling

To identify proteins proximal to pTF, quintuplicate cultures from one WT and one pTF-APEX2 expressing cell line (cell line s2, *Figure 2C–D*) were grown to mid- to late-exponential phase and supplemented with 2.5 mM biotin-phenol and 1 mM hydrogen peroxide, following the APEX2 proximity labeling protocol developed for yeast (*Hwang and Espenshade, 2016*; *Hwang et al., 2016*). These labeling reaction steps were performed at 4°C and the incubation with hydrogen peroxide was extended to 20 min to mirror Amplex UltraRed assay and DAB reaction conditions (*Figure 4A*). Increasing biotin-phenol concentration from 0.5 mM, usually used in mammalian cells, to 2.5 mM, and exposing cells to osmotic stress with 1.2 M sorbitol—both of which are critical for efficient labeling in yeast (*Hwang and Espenshade, 2016*)—was necessary to detect enrichment of biotinylated proteins in experimental samples (*Figure 4B*). This result confirmed our hypothesis that the lack of

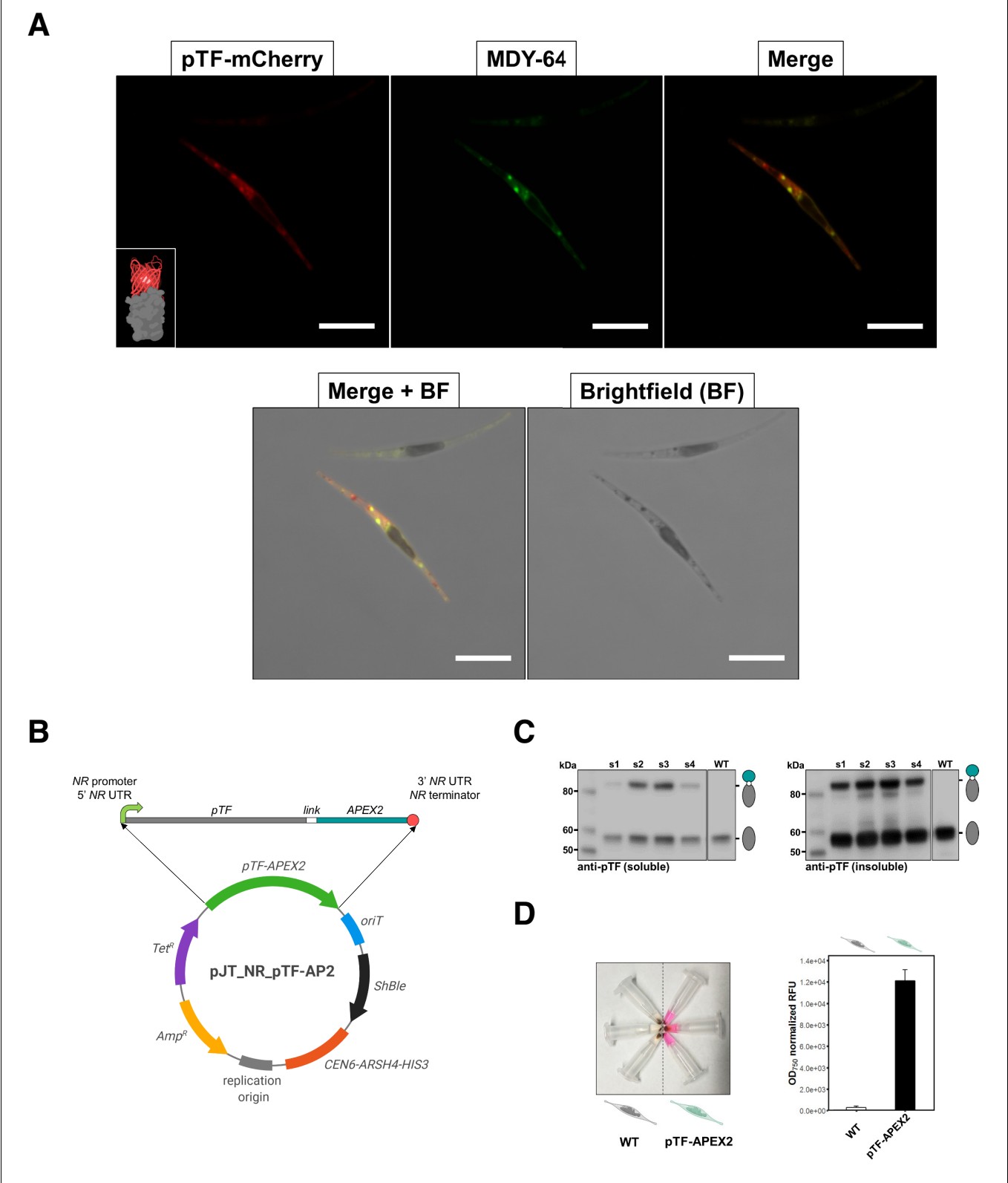

**Figure 2.** pTF is localized to intracellular vesicles and APEX2 is enzymatically active in live *Phaeodactylum tricornutum* cells. (**A**) pTF-mCherry colocalizes with the membrane dye MDY-64 on the cell surface and within intracellular vesicles. The fluorescent fusion protein was expressed under the *pTF* promoter and terminator in a Δ*pTF P. tricornutum* genetic background. Cells were stained with 100 µM MDY-64 for 10 min at room temperature. Scale bar is 10 µm. Fusion protein schematic created with BioRender.com. (**B**) Top: *pTF-APEX2* flanked by nitrate-inducible *NR* (nitrate reductase) promoter,

*Figure 2 continued on next page*

*Figure 2 continued*

terminator, and untranslated regions (UTRs). Linker sequence ('*link*') between *pTF* and *APEX2* encoded KGSGSTSGSG. Bottom: Schematic of the pTF-APEX2 expressing episome. Tetracycline (*Tet^R*) and ampicillin (*Amp^R*) resistance genes for *E. coli* selection, bacterial replication origin, yeast centromere (*CEN6-ARSH4-HIS3*) for episome maintenance, phleomycin/zeocin (*ShBle*) resistance gene for *P. tricornutum* selection, origin of conjugative transfer (*oriT*). Plasmid map created with BioRender.com. (**C**) Anti-pTF western blot confirming pTF-APEX2 fusion protein (84.73 kDa) expression in four transconjugant *P. tricornutum* cell lines (lanes s1–s4; left: soluble cell lysate fraction, right: insoluble cell lysate fraction). Bacterial conjugation was performed with WT *P. tricornutum* cells which is why native pTF (57.05 kDa) bands were present in our samples. Teal circle: APEX2, white circle: linker, gray oval: pTF. (**D**) Left: pTF-APEX2 expressing, but not WT, cells convert APEX2 substrate Amplex UltraRed (50 µM) into a colored product resorufin in the presence of 2 mM H$_2$O$_2$. Right: >40 fold higher resorufin signal was observed in supernatants from pTF-APEX2 expressing cells than those from WT cells. Triplicate cultures from one WT and one pTF-APEX2 expressing cell line (cell line s2) were used in this experiment. Standard deviation is shown. *P. tricornutum* cartoons created with BioRender.com.

The online version of this article includes the following source data and figure supplement(s) for figure 2:

**Source data 1.** Raw Amplex UltraRed assay data underpinning the chart in *Figure 2D*.

**Source data 2.** R script to reproduce the chart in *Figure 2D*.

**Source data 3.** CSV file to be used in conjuction with *Figure 2—source data 2*.

**Figure supplement 1.** *Phaeodactylum tricornutum* phytotransferrin (pTF) features, resorufin imaging in pTF-APEX2 expressing cells, and resorufin signal co-dependence on Amplex UltraRed and hydrogen peroxide.

heavy silicification in *P. tricornutum* (*Francius et al., 2008*; *Tesson et al., 2009*), making its cell wall composition and cell membrane permeability likely similar to that of yeast, would permit labeling. Streptavidin pull-downs were then performed with clarified cell lysates followed by tandem mass tag (TMT)-based quantitative proteomics. WT and pTF-APEX2 proteomic replicates, with the exception of one WT and one pTF-APEX2 sample, formed two distinct clusters (*Figure 4—figure supplement 1*; *Figure 4—figure supplement 1—source data 1*) indicating minimal technical variability. Thirty-eight statistically significant proteins (p value ≤ 0.05) with APEX2/WT ratios of at least 1.5 were identified (*Figure 4B*; *Figure 4—source data 1*). These ratios were obtained from average total peptide counts across quintuplicates (*Figure 4—source data 1*). Endogenous biotinylated proteins were also detected and had APEX2/WT ratios close to one, thus acting as an intrinsic pull-down control (*Supplementary file 1*—Table S1; *Figure 4—source data 1*). Some background enrichment in WT cells was likely due to endogenous APEX2-like peroxidases and one would therefore expect mitochondrial proteins with APEX2/WT ratios close to one to be present in our MS dataset. Indeed, at least three were detected: mitochondrial chaperonin CPN60 (Phatr3_J24820, UniProt ID: B7FQ72, APEX2/WT = 0.99), mitochondrial import receptor subunit TOM70 (Phatr3_J47492, UniProt ID: B7G3J4, APEX2/WT = 0.91), and acetyl-CoA dehydrogenase (Phatr3_J11014, UniProt ID: B7FTR6, APEX2/WT = 1.23) (*Figure 4—source data 1*). Fourteen proteins with an APEX2/WT ratio of at least two were detected. Of these 14 proteins, nine are known to be transcriptionally sensitive to iron availability; this is also the case for an additional five proteins with APEX2/WT ratio of at least 1.5 (*Figure 5—figure supplement 1—source data 1*; *Smith et al., 2016*). We note that pTF was present, but not enriched, in our pTF-APEX2 proteomics samples (*Figure 4—source data 1*). One possible explanation for this result is that pTF may not have surface exposed amino acid residues that are permissive to biotinylation by APEX2-generated phenoxyl radicals and so its presence in all of our pull-down samples could be due to unspecific binding to streptavidin beads.

Some of the most apparent biologically interesting hits are summarized in *Figure 4C*. Considering iron is critically involved in all of the major photosynthetic complexes—photosystem II, photosystem I, and cytochrome *b6f* (*Rochaix, 2011*)—we asked if some of the proteins are perhaps predicted to be targeted to the chloroplast. Indeed, five are, albeit with low confidence, by ASAFind (*Gruber et al., 2015*), including two (Phatr3_J51183 and Phatr3_J54986) that are part of a gene cluster on chromosome 20 and two (Phatr3_J41423 and Phatr3_J55031) that are known to associate with the *P. tricornutum* chloroplast (*Figure 4C*; *Allen et al., 2012*; *Kazamia et al., 2019*; *Kazamia et al., 2018*). We elaborate on the potential role for these proteins in intracellular iron trafficking in the Discussion.

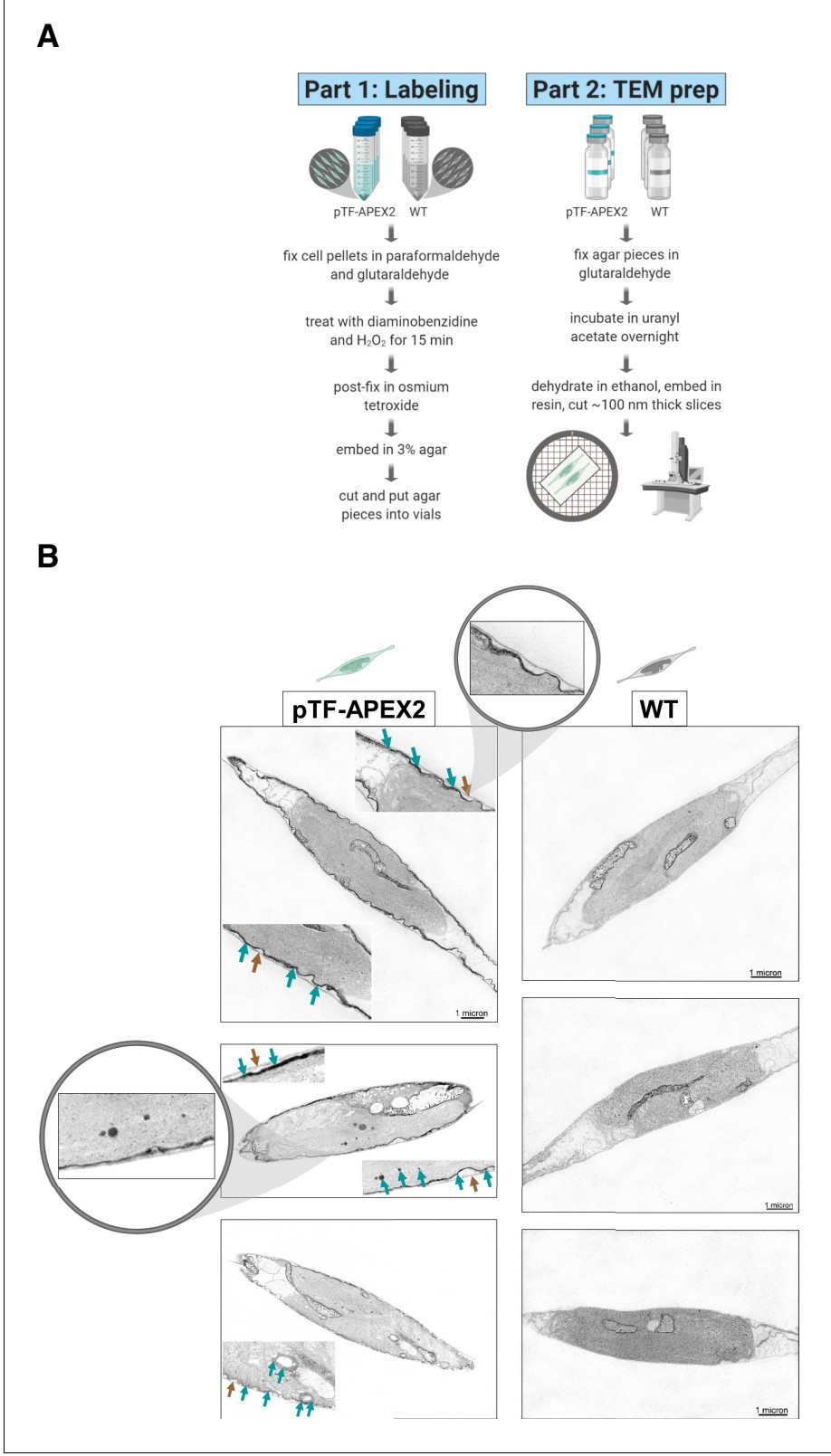

**Figure 3.** pTF-APEX2 is localized to the cell membrane and intracellular vesicles. (A) Electron microscopy protocol summary. Briefly: WT and pTF-APEX2 expressing cells (grown in triplicate) were fixed, treated with diaminobenzidine (DAB) and hydrogen peroxide, post-fixed with osmium tetroxide, embedded in agar, negatively stained with uranyl acetate, dehydrated, embedded in resin, and visualized. Created with BioRender.com. (B) Expected cell surface and intracellular pTF-APEX2 localization. Teal and brown arrows point to APEX2-induced signal and cell wall, respectively. Zoomed in (top

*Figure 3 continued on next page*

*Figure 3 continued*

left image) is cell periphery where it can clearly be seen that cell membrane, not cell wall, is occupied by the fusion protein. pTF-APEX2 containing vesicles were observed (bottom two left images; acquired with backscatter scanning electron microscopy). Zoomed in (middle left image) are cell membrane and vesicles with pTF-APEX2. Mitochondrial signal in both WT and pTF-APEX2 expressing cells is likely due to endogenous (mitochondrial) APEX2-like peroxidases. Scale bar is 1 μm. *P. tricornutum* cartoons created with BioRender.com.

The online version of this article includes the following figure supplement(s) for figure 3:

**Figure supplement 1.** Analysis of transcriptionally active *Phaeodactylum tricornutum* peroxidases reveals putative APEX2-like and mitochondrial enzymes.

## Proteins encoded by an iron-sensitive gene cluster on chromosome 20 colocalize with pTF

We focused on three proteins with APEX2/WT ratio of at least two—Phatr3 IDs: J51183 (hereafter pTF.CREG1), J52498 (hereafter pTF.CatCh1), and J54986 (hereafter pTF.ap1)—that are expressed from a previously identified iron- and silicon-sensitive gene cluster on chromosome 20 (*Figure 5A*; *Allen et al., 2008*; *Sapriel et al., 2009*). One additional protein, ISIP2b (Phatr3_J54987, UniProt ID: B7G9B1), which is also transcriptionally sensitive to iron and silicon and that we did not detect in our proximity proteomics experiment, is also encoded by this uncharacterized locus. We note that *pTF* is not co-located with these genes, but instead lies on chromosome 7 (genomic location 1,000,053–1,001,833, forward strand). All three genes exhibit a transcriptional profile similar to *pTF* (*Figure 5—figure supplement 1*), two proteins (pTF.CREG1 and pTF.ap1) are predicted to go to the chloroplast (with low confidence as determined by ASAFind), and two (pTF.CatCh1 and pTF. ap1) contain a C-terminal transmembrane domain (*Figure 4C*; *Supplementary file 1*—Table S2). To test whether they colocalize with pTF, co-expression episomes were assembled and conjugated into WT *P. tricornutum* cells which resulted in diatom cell lines expressing pTF-mCherry and MS hit-EYFP fusion proteins (*Figure 5—figure supplement 2A–B*). Imaging conditions were optimized with mCherry and Venus (yellow fluorescent protein with spectral properties very similar to those of EYFP) expressing *P. tricornutum* cell lines for minimal cross-channel bleed-through (*Figure 5—figure supplement 2C*). pTF.CREG1 and pTF.CatCh1 colocalized with pTF in the chloroplast vicinity and on the chloroplast margin, respectively (*Figure 5B*). Both proteins consistently exhibited these distinct colocalization patterns as demonstrated by additional three mCherry- and EYFP-positive cells from corresponding co-expression transconjugant cell lines (*Figure 5—figure supplement 3A–B*). Colocalization of pTF.ap1 with pTF close to the chloroplast was also evident, although somewhat less precise (*Figure 5—figure supplement 4*).

## pTF.CREG1 has homologs across the tree of life and may possess a non-enzymatic role similar to human CREG1

To shed light on possible functions of the uncharacterized proteins colocalizing with pTF and to examine their occurrence in other diatoms and marine phytoplankton beyond *Phaeodactylum tricornutum*, phylogenetic analysis was performed. pTF.CREG1 homologs were identified across the tree of life (*Figure 6*; *Figure 6—source data 1*). Further investigation of the protein sequence alignment underlying the phylogenetic tree revealed 12 amino acid residues that are at least 90% conserved across all homologs (*Figure 6—figure supplement 1*). Diatom homologs can be seen in a crown group with proteins from other complex plastid-containing algae such as cryptophytes, haptophytes, pelagophytes, chlorarachniophytes, and dinoflagellates (*Figure 6*; *Archibald, 2009*; *Füssy and Oborník, 2018*). pTF.CREG1 and its iron-insensitive paralog Phatr3_J10972—which phylogenetically cluster together—have ~50% amino acid sequence similarity and belong to the flavin mononucleotide (FMN)-binding split barrel fold protein superfamily (SSF50475) according to their Ensembl Protists profiles.

We analyzed pTF.CREG1 amino acid sequence (UniProt ID: B7G9B3, 234 AAs, ~26.4 kDa) and compared it to the human CREG1 ortholog (UniProt ID: O75629, 220 AAs, ~24 kDa). While human CREG1 is evolutionarily and structurally closely related to flavin mononucleotide (FMN)-coordinating dimeric oxidases and oxidoreductases such as pyridoxine 5'-phosphate (PNP) oxidases, it is unable to bind FMN and catalyze NADPH oxidation (*Ghobrial et al., 2018*; *Sacher et al., 2005*). pTF. CREG1 has a predicted N-terminal signal peptide ending in SGA that may be recognized by an

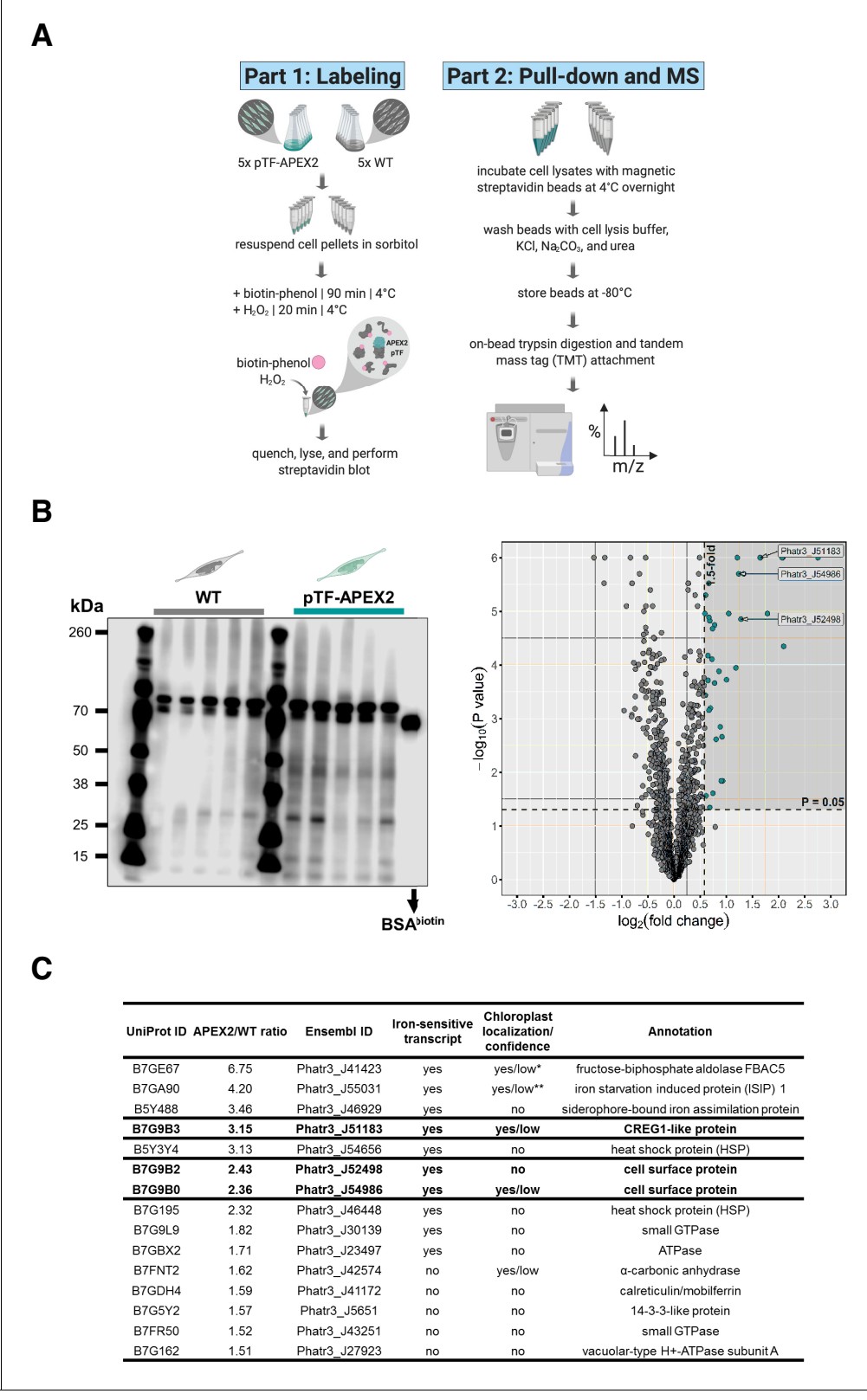

**Figure 4.** Proximity-dependent proteomic mapping with APEX2 reveals candidate proteins involved in phytotransferrin (pTF) endocytosis in *Phaeodactylum tricornutum*. (**A**) Summary of the proximity-dependent proteomic mapping experiment. Briefly: WT and pTF-APEX2 expressing cells (grown in quintuplicate) were chilled on ice, pelleted, treated with 1.2 M sorbitol, supplemented with 2.5 mM biotin-phenol and 1 mM $H_2O_2$. The labeling reaction was quenched, cells were lysed, and evaluated for biotin enrichment. Cleared cell lysates were then subjected to streptavidin pull-
*Figure 4 continued on next page*

*Figure 4 continued*

down followed by quantitative mass spectrometry using tandem mass tags (TMT). Created with BioRender.com. (B) Left: Enrichment of biotinylated proteins over WT background was observed with streptavidin blot; ~66.5 kDa biotinylated BSA control; equal loading. The most prominent band in all samples is likely the endogenous biotin-containing propionyl-CoA carboxylase (Phatr3_J51245, ~72.5 kDa). Right: Volcano plot of quantitative MS data highlighting proteins with APEX2/WT ratio of at least 1.5 and p value ≤ 0.05 (shaded area with teal data points). Proteins encoded by a known iron-sensitive gene cluster on chromosome 20 are highlighted (arrowed Ensembl IDs). *P. tricornutum* cartoons created with BioRender.com. (C) 14/38 (10 shown here) MS hits are proteins with iron-sensitive transcripts. Proteins chosen for further characterization are bolded. Chloroplast localization and the associated prediction confidence were determined with SignalP 4.1 and ASAFind version 1.1.7. *Experimentally shown to localize to the pyrenoid—a RuBisCO-containing proteinaceus organelle—in the interior of the *P. tricornutum* chloroplast (*Allen et al., 2012*). **Experimentally shown to be localized adjacently to the chloroplast (*Kazamia et al., 2019*; *Kazamia et al., 2018*).

The online version of this article includes the following source data and figure supplement(s) for figure 4:

**Source data 1.** Pre-processed quantitative mass spectrometry proteomics data underpinning the Volcano plot and the table in *Figure 4B* and *Figure 4C*, respectively.

**Source data 2.** R script to reproduce the chart in *Figure 4B*.

**Source data 3.** CSV file to be used in conjuction with *Figure 4—source data 2*.

**Figure supplement 1.** WT and pTF-APEX2 proteomic replicates form distinct clusters.

**Figure supplement 1—source data 1.** Scaled quantitative mass spectrometry proteomics data.

---

endoplasmic reticulum-localized signal peptidase (*Benham, 2012*; *Tuteja, 2005*), and is likely modified by both N- and O-glycosylation, in line with human CREG1 (*Figure 6—figure supplement 2A*; *Sacher et al., 2005*). These features suggest that pTF.CREG1 enters and travels along the secretory pathway (*Bard and Chia, 2016*; *Benham, 2012*). Furthermore, the amino acid sequence alignment between pTF.CREG1 and human CREG1 revealed a similar tetrapeptide motif DP(Q/E)S which is located on the loop occluding the FMN-binding pocket at the human CREG1 dimer interface (*Figure 6—figure supplement 1*; *Figure 6—figure supplement 2A*; *Sacher et al., 2005*). This short amino acid sequence—while thought to contribute to the lack of catalytic activity in human CREG1—was found to be critical for mediating cell growth inhibition in human teratocarcinoma cell line NTERA-2 (*Sacher et al., 2005*). Structural comparison between a pTF.CREG1 homology model created using Phyre2 web portal (*Kelley et al., 2015*) and the human CREG1 monomer (PDB ID: 1XHN; *Sacher et al., 2005*) supports our hypothesis that pTF.CREG1 also contains a loop preventing FMN binding (*Figure 6—figure supplement 2B*). Two arginine amino acid residues involved in binding the terminal FMN phosphate in *S. cerevisiae* pyridoxine 5′-phosphate oxidase (UniProt ID: P38075, PDB ID: 1CI0) are mutated to aspartic acid and methionine amino acid residues in human CREG1 (*Sacher et al., 2005*), and one of these two key arginine residues is absent from pTF.CREG1 (*Figure 6—figure supplement 3A*). Finally, we note that a putative homodimerization region in pTF.CREG1 homology model contains many hydrophobic amino acid residues and overall follows the compositional character of the human CREG1 dimerization interface suggesting pTF.CREG1 is acting as a dimer in vivo (*Figure 6—figure supplement 3B*).

Given the apparent structural similarity of the *P. tricornutum* pTF.CREG1 to the catalytically inactive human ortholog, we opted to test its activity, or lack thereof, to delineate it from the evolutionarily related FMN-binding enzymes. Various truncations of His6-tagged pTF.CREG1 were expressed in *Escherichia coli* and purified using cobalt, $Co^{2+}$, immobilized metal affinity chromatography (IMAC) to test solubility. A construct encoding pTF.CREG1 without the 31 N-terminal amino acid residues and leading to a soluble protein was selected for larger-scale purification (*Figure 6—figure supplement 4A*). The $Co^{2+}$-bound protein fraction (*Figure 6—figure supplement 4B*) exhibited no flavin reductase activity as measured by the oxidation of NADPH in the presence of potential flavin substrates flavin mononucleotide (FMN) and riboflavin (*Figure 6—figure supplement 2C*; *Figure 6—figure supplement 2—source data 1*).

In conclusion, our homology modeling analyses and enzymatic assays indicate that pTF.CREG1 does not bind FMN and does not reduce flavins in an NADPH-dependent manner, respectively, both in agreement with our current knowledge about human CREG1. As such, pTF.CREG1 is an enticing candidate for further study as a key participant in the endocytic pTF receptor-mediated iron uptake pathway in diatoms. We put forth a hypothesis for its function in the Discussion.

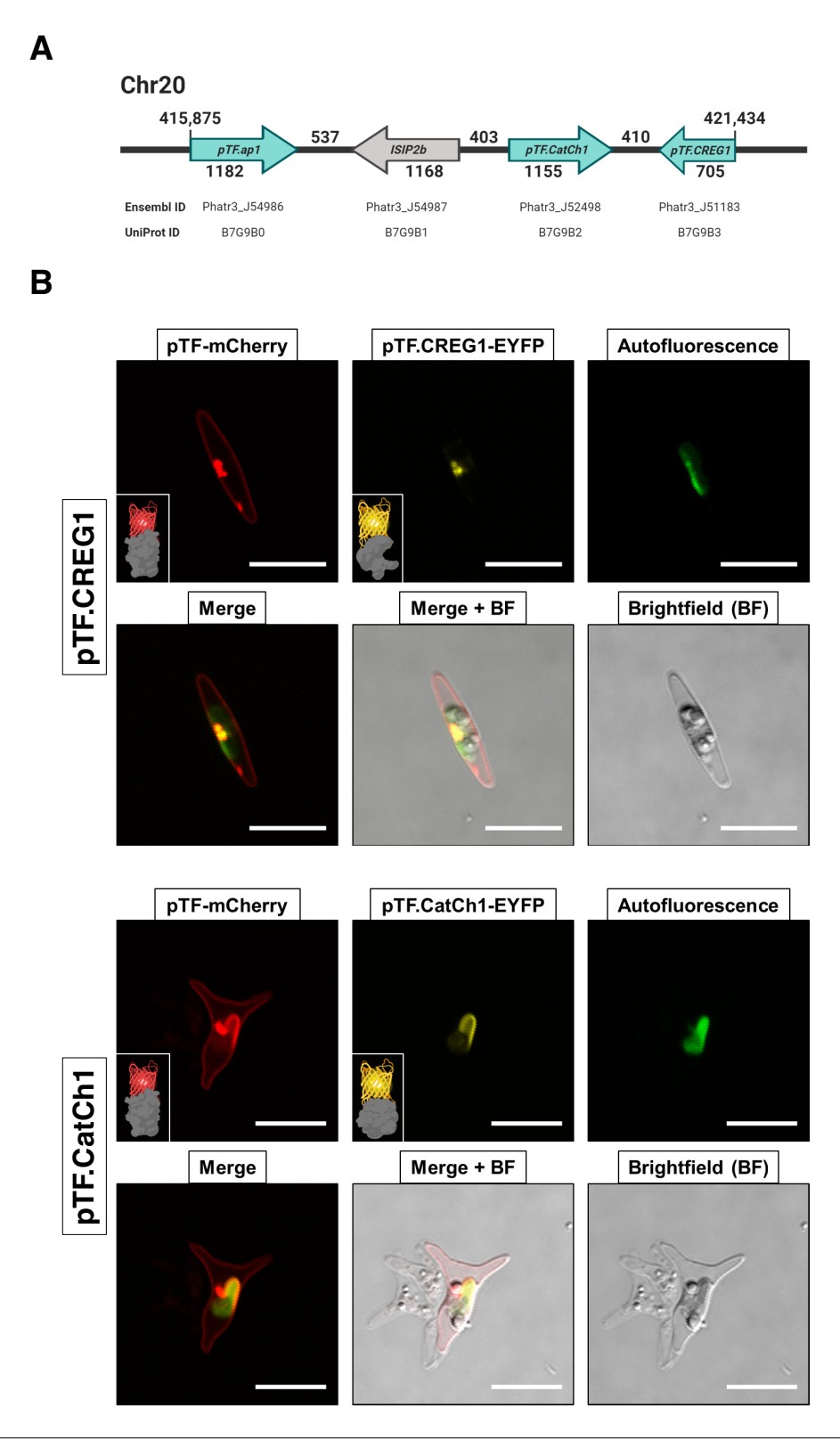

**Figure 5.** Proteins encoded by a known iron- and silicon-sensitive locus in *Phaeodactylum tricornutum* colocalize with pTF in the chloroplast vicinity. (**A**) Genes corresponding to three statistically significant proteomic hits are clustered on chromosome 20. All three proteins were co-expressed with mCherry-tagged pTF as fusions with a yellow fluorescent protein. Corresponding Ensembl and UniProt IDs are noted. Numbers indicate base pairs. Created with BioRender.com. (**B**) Two proteins—pTF.CREG1 and pTF.CatCh1—show clear, yet distinct, colocalization with pTF and the chloroplast

*Figure 5 continued on next page*

Figure 5 continued

periphery. pTF.CREG1-EYFP was consistently punctate whereas pTF.CatCh1-EYFP lined the chloroplast margin. pTF-mCherry punctum was almost exclusively positioned next to this 'chloroplast lining' pattern. *P. tricornutum* is pleiomorphic which explains why different cell morphologies were observed in pTF.CREG1-EYFP (fusiform morphotype) and pTF.CatCh1-EYFP (triradiate morphotype) cell lines. Scale bar is 10 μm. Fusion protein schematics created with BioRender.com.

The online version of this article includes the following source data and figure supplement(s) for figure 5:

**Figure supplement 1.** *pTF.CREG1, pTF.CatCh1,* and *pTF.ap1* closely resemble *pTF* transcriptional profile.

**Figure supplement 1—source data 1.** Transcriptomic data and functional annotations for proteins with APEX2/WT ratios $\geq$ 1.5 and p values $\leq$ 0.05 underpinning *Figure 5—figure supplement 1* and *Figure 4C*.

**Figure supplement 1—source data 2.** R script to reproduce the chart in *Figure 5—figure supplement 1*.

**Figure supplement 1—source data 3.** CSV file to be used in conjuction with *Figure 5—figure supplement 1—source data 2*.

**Figure supplement 2.** Co-expression episome assembly overview, assessment of co-expression cell lines, and confocal microscopy bleed-through controls.

**Figure supplement 3.** Additional representative examples of pTF.CREG1 and pTF.CatCh1 colocalization with pTF.

**Figure supplement 4.** pTF.ap1 colocalizes with pTF.

## pTF.CatCh1 is a heterokont-specific protein with conserved CX(X)C motifs

In contrast to pTF.CREG1, pTF.CatCh1 homologs were identified only in diatoms and other single-celled heterokonts (*Figure 7A*; *Figure 7—source data 1*). pTF.CatCh1 is a ~37.4 kDa (349 AAs) large protein with a predicted signal peptide (AAs 1–26) ending in ASA, two N-glycosylation NX(S/T) sequons (i.e. asparagines in the identified NST and NLS motifs may be glycosylated), and 13 predicted O-glycosylation sites (data not shown; *Rao and Bernd, 2010*). This protein is paralogous to the iron starvation induced protein 2b (ISIP2b/Phatr3_J54987) whose gene lies adjacently to *pTF.CatCh1* in the same chromosome 20 gene cluster (*Figure 5A*). They share three CXC and one CXXC motifs typical of metal-binding, redox-active, and iron–sulfur cluster-containing proteins (*Figure 7B*; *Blaby-Haas et al., 2014*; *Fomenko and Gladyshev, 2003*; *Poole, 2015*; *Przybyla-Toscano et al., 2018*). One of the CXC motifs and the CXXC motif are conserved across all but two homologs in our phylogenetic analysis (*Figure 7B*; *Figure 7—figure supplement 1A*), and the latter is predicted to be located in one of the two disordered C-terminal regions in pTF.CatCh1 (AAs 196–292, AAs 321–349; *Figure 7—figure supplement 1B*). With additional 12 cysteines conserved between pTF.CatCh1 and ISIP2b, pTF.CatCh1 contains a total of 20 cysteine amino acid residues (all between AAs 34 and 226). Six of these 20 are 100% conserved across all homologs (*Figure 7B*). The predicted transmembrane domain (AAs 296–315) is flanked by a polyserine ([Ser]$_6$) and a short arginine-rich stretch (RKL[R]$_3$). Flexible polyserine linkers are found in modular, multidomain proteins (*Uversky, 2015a*), and positively charged amino acid residue tracts in the vicinity of transmembrane regions were shown to be orientation determinants in outer chloroplast membrane proteins (*May and Soll, 1998*). These observations are synthesized in a domain organization schematic shown in *Figure 7—figure supplement 1C*. We elaborate on possible roles of the conserved CX(X)C motifs, as well as other identified motifs and domains in pTF.CatCh1 in the Discussion.

## pTF.ap1 homologs are present in marine microeukaryotes including diatoms

Phylogenetic characterization was performed with pTF.ap1 as well and numerous homologs with at least five highly conserved motifs were identified in diatoms and other marine microeukaryotes, including chlorophytes, cryptophytes, and haptophytes (*Figure 7—figure supplement 2A–B*; *Figure 7—figure supplement 2—source data 1*). These pTF.ap1 homologs contain five 100% conserved cysteine amino acid residues, two of which are shown in *Figure 7—figure supplement 2B*, as well as 100% conserved tyrosine, histidine, and aspartic acid residues, indicating this protein may also be redox-active and/or have a metal ion processing role (*Dokmanić et al., 2008*).

## Discussion

In this study, we used the iron receptor protein phytotransferrin (pTF) to implement APEX2-mediated proximity-dependent proteomic mapping in the model marine diatom *Phaeodactylum*

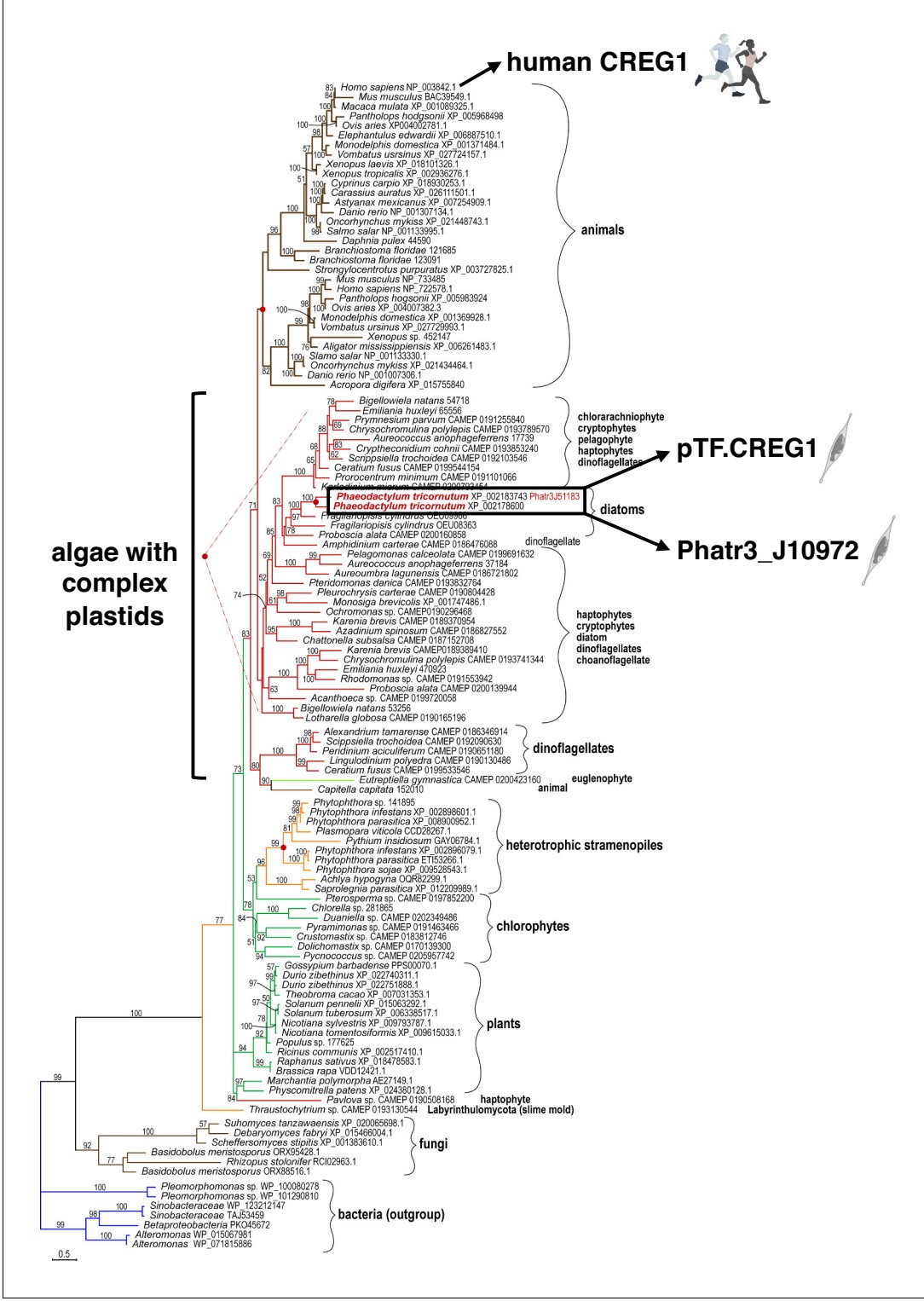

**Figure 6.** pTF.CREG1 homologs are present across the tree of life. pTF.CREG1 homolog search was performed with the National Center for Biotechnology Information (NCBI) and the Marine Microbial Eukaryote Transcriptome Sequencing Project (MMETSP) databases (*Caron et al., 2017*; *Keeling et al., 2014*). Homologs from algae with complex plastids (including diatoms) form a crown group away from animal proteins. pTF.CREG1 clusters with its paralog Phatr3_J10972 (black rectangle). Red dots indicate predicted gene duplication events. Scale bar: 0.5 substitutions per position.

The online version of this article includes the following source data and figure supplement(s) for figure 6:

*Figure 6 continued on next page*

*Figure 6 continued*

**Source data 1.** Amino acid sequence alignment underpinning the phylogenetic tree in *Figure 6*.

**Figure supplement 1.** Conserved amino acid residues across pTF.CREG1 homologs.

**Figure supplement 2.** Bioinformatic, structural, and enzymatic insights suggest pTF.CREG1 has a non-catalytic function similar to human CREG1.

**Figure supplement 2—source data 1.** Flavin reduction enzymatic assay data underpinning charts in *Figure 6— figure supplement 2C*.

**Figure supplement 3.** pTF.CREG1 is unlikely able to coordinate the terminal FMN phosphate and may act as a dimer in vivo.

**Figure supplement 4.** pTF.CREG1 expression and purification.

---

*tricornutum*. The resulting data provide several advances regarding the function of pTF and its putative protein interaction partners downstream of iron binding at the cell surface.

pTF (ISIP2a; iron starvation induced protein 2a) was identified in *P. tricornutum* as a marker for iron (Fe) limitation (*Allen et al., 2008*). It was subsequently shown to be involved in Fe acquisition (*Morrissey et al., 2015*) and to be widespread in marine and freshwater microeukaryotic phytoplankton communities (*Bertrand et al., 2015*; *McQuaid et al., 2018*; *Tara Oceans Coordinators et al., 2018*). In a breakthrough study, *McQuaid et al., 2018*, demonstrated that ISIP2a is a type of transferrin, phytotransferrin (pTF), that is crucial for acquisition of dissolved labile ferric, $Fe^{3+}$, iron, a critical micronutrient for cellular biochemistry. *McQuaid et al., 2018*, showed that pTF is essential for high-affinity iron uptake in *P. tricornutum*, and that carbonate, $CO_3^{2-}$, and $Fe^{3+}$ interact synergistically to control the iron uptake rate. The finding that carbonate anions are required for the activity of a key diatom iron transport system is noteworthy and suggests that ocean acidification might inhibit microbial iron uptake. The occurrence of transferrin in marine diatoms and other single-celled marine and freshwater phytoplankton raises significant questions concerning its evolutionary origin and cellular role.

*McQuaid et al., 2018* observed pTF in intracellular puncta, showed that the clathrin-mediated endocytosis inhibitor Pitstop 2 reduces iron (Fe) uptake rates, visualized vesicles after adding iron to an iron-limited WT *P. tricornutum* culture stained with the membrane dye FM 1–43, but our MDY-64 labeling data provide the first direct evidence that diatom pTF is indeed associated with intracellular membranous compartments. Further work is needed to elucidate pTF dynamic upon cellular internalization—especially after iron addition to iron-deplete cells which mimics oceanic iron repletion events—and its trafficking in and potential exit from the endomembrane system in relation to cell cycle phases and light-dark cycles (*Mayle et al., 2012*; *Naslavsky and Caplan, 2018*; *Smith et al., 2016*).

While APEX2 was expected to be active in *P. tricornutum* given previous heterologous expression of functional horseradish peroxidase (HRP) in the model marine diatom *Thalassiosira pseudonana* (*Sheppard et al., 2012*), it was unclear whether endogenous peroxidases would contribute to the background signal in various APEX2 assays. In our Amplex UltraRed assay experiments, an order of magnitude higher signal-to-noise ratio (i.e. resurofin signal in pTF-APEX2 expressing versus WT *P. tricornutum* cells) was detected when live cells were reacted on ice as opposed to on room temperature, suggesting endogenous *P. tricornutum* peroxidases, but not exogenous APEX2, are largely inhibited on ice. This is notable as it indicates that performing APEX2 assays on ice may represent a generalizable experimental strategy for mesophilic microeukaryotic phytoplankton model systems.

Identifying Amplex UltraRed assay conditions which minimized WT background was promising, but not a guarantee that some endogenous enzymes would not be able to process other APEX2 substrates, such as those crucial for electron microscopy (3,3'-diaminobenzidine/DAB) and proximity proteomics (biotin-phenol). Indeed, mitochondrial background signal was detected in our electron microscopy experiments both in WT and pTF-APEX2 expressing cells, and our *P. tricornutum* proteome analysis revealed candidate endogenous APEX2-like peroxidases that could explain this result. Nevertheless, C-terminal tagging of pTF with APEX2 resulted in the expected cell surface and what appeared to be vesicular fusion protein localization away from mitochondria indicating no spatial overlap between endogenous APEX2-like enzymes and pTF-APEX2. Inconsistent size and shape of vesicle-like structures observed in our electron micrographs were conceivably due to a combination of arbitrary cell orientation inside agar blocks (where diatom cells were embedded) and the exact z

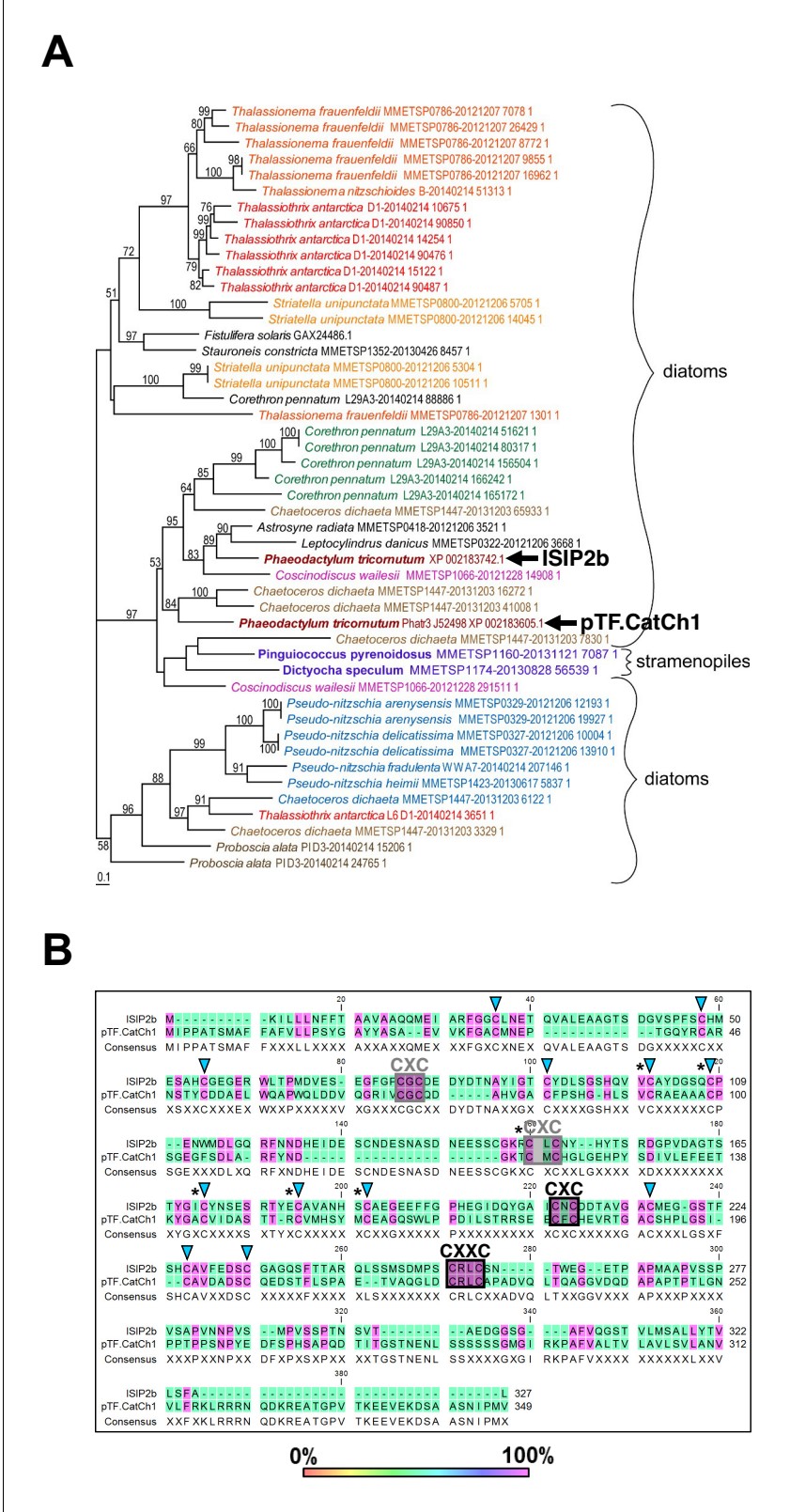

**Figure 7.** pTF.CatCh1 is a cysteine-rich heterokont-specific protein. (**A**) pTF.CatCh1 homolog search was performed with NCBI and marine microbial eukaryote (MMETSP) databases. Numerous, almost exclusively diatom, proteins were identified. Interestingly, pTF.CatCh1 is paralogous to ISIP2b (black arrows). Scale bar: 0.1 substitutions per position. (**B**) pTF.CatCh1 and ISIP2b share three CXC and one CXXC motifs (boxed) commonly

*Figure 7 continued on next page*

*Figure 7 continued*
found in redox-active proteins and proteins involved in cellular metal homeostasis. Black boxes indicate the two CX(X)C motifs conserved across the majority of pTF.CatCh1 homologs. Blue triangles: additional conserved cysteine amino acid residues between pTF.CatCh1 and ISIP2b for a total of 20. Asterisks: cysteine amino acid residues that are 100% conserved across all homologs.

The online version of this article includes the following source data and figure supplement(s) for figure 7:

**Source data 1.** Amino acid sequence alignment underpinning the phylogenetic tree in *Figure 7A*.
**Figure supplement 1.** Conservation of CX(X)C motifs in pTF.CatCh1 homologs and pTF.CatCh1 features.
**Figure supplement 2.** pTF.ap1 phylogeny and conserved motifs.
**Figure supplement 2—source data 1.** Amino acid sequence alignment underpinning the phylogenetic tree in *Figure 7—figure supplement 2A*.

position of ultramicrotome cut sites. On the whole, APEX2-based imaging is complementary to super-resolution microscopy in diatoms and represents a basis for electron tomography applications (*Gröger et al., 2016*; *Sengupta et al., 2019*). Some off-target biotinylation in our proximity labeling experiments was likely due to background mitochondrial peroxidase activity as indicated by the presence of mitochondrial proteins in our mass spectrometry data. Their APEX2/WT ratios close to one give us more confidence that those with elevated ratios above 1.5 were enriched due to APEX2 activity. Many of the latter have functional annotations for proteins we might expect to play a role in endosomal iron assimilation, and many have previously been shown to have iron-sensitive transcripts, both of which further support this notion. Future efforts to express pTF tagged with N-terminal APEX2, which should lead to intravesicular proximity labeling according to our pTF orientation prediction, optimizing conditions for APEX2 reactivation in iron-deplete conditions with hemin chloride, and more cell wall-permeant APEX2 probes (*Li et al., 2020*) will allow for identification of additional candidate proteins. Considering iron metabolism genes in *P. tricornutum*, including *pTF*, are highly expressed at night, proteomic maps with improved spatial and temporal resolution may emerge from proximity-dependent proteomic mapping experiments conducted with synchronized cell lines sampled at different timepoints throughout diel cycle (*Huysman et al., 2014*; *Lundberg and Borner, 2019*; *Smith et al., 2016*). The successful implementation of TurboID—an engineered biotin ligase for proximity proteomics (*Branon et al., 2018*)—in model plants *Arabidopsis thaliana* (*Kim et al., 2019*; *Mair et al., 2019*) and *Nicotiana benthamiana* (*Zhang et al., 2019*) suggests its adoption in existing and emerging single-celled marine phytoplankton model systems, including diatoms, is likely (*Faktorová et al., 2020*; *Falciatore et al., 2020*). TurboID is an iron-independent protein and could thus be particularly valuable to further advance iron metabolism studies in *P. tricornutum*. These proposed improvements and alternative experimental approaches should cross-validate the proteins identified in this study as well as illuminate new ones.

Functionally uncharacterized gene cluster on chromosome 20 we zoomed in on is transcriptionally upregulated in iron-deplete (*Allen et al., 2008*; *Smith et al., 2016*) and silicic-acid-replete (*Sapriel et al., 2009*) conditions corroborating a known link between iron and silicon metabolism in diatoms (*Brzezinski et al., 2015*; *Durkin et al., 2012*; *Hutchins and Bruland, 1998*; *Leynaert et al., 2004*). Physically clustered functionally related genes are a hallmark of prokaryotic biology and operon-like genomic elements have been described in fungi and plants (*Nützmann et al., 2018*; *Osbourn and Field, 2009*); however, identifying and characterizing them in marine diatom genomes is in its relative infancy. For example, genes encoding light harvesting fucoxanthin-chlorophyll-binding proteins (FCPs) are co-located on chromosome 2 in *P. tricornutum* (*Bhaya and Grossman, 1993*), and a compact ~7 kbp gene cluster encoding enzymes for domoic acid neurotoxin production was characterized in a cosmopolitan diatom *Pseudo-nitzschia multiseries* (*Brunson et al., 2018*). Given that diatom genomes have likely been extensively shaped by horizontal gene transfer (*Bowler et al., 2008*; *Diner et al., 2017*), it is interesting to note that some yeast species acquired iron acquisition-enabling operon-like genomic elements via horizontal operon transfer (HOT) from an ancient bacterium (*Kominek et al., 2019*). Our phylogenetic analyses suggest all three proteins—pTF.CREG1, pTF.CatCh1, pTF.ap1—encoded by this locus are present in diatoms and other (marine) microeukaryotes, while pTF.CREG1 homologs are found across the tree of life. They, alongside the rest of our proteomic data, paint an emerging view of intracellular ferric, $Fe^{3+}$, iron processing events downstream of its phytotransferrin-mediated binding at the cell surface.

In humans, endosome acidification causes structural rearrangements in the transferrin protein and protonation of synergistically coordinated carbonate anion leading to ferric iron release (*Cheng et al., 2004*). This event is concurrent with or followed by endosomal reductase STEAP3-catalyzed ferric iron reduction and ferrous, $Fe^{2+}$, iron export through divalent metal transporter DMT1/NRAMP2 or Zrt- and Irt-like protein (ZIP) 14 (*Bogdan et al., 2016*). Two putative proton, $H^+$, pumps were present in our mass spectrometry data: Phatr3_J23497—predicted as a P-type ATPase—and Phatr3_J27923—a vacuolar-type $H^+$-ATPase subunit A. Multisubunit V-ATPases regulate pH homeostasis in virtually all eukaryotes and are known to function as endosome acidifying protein complexes (*Finbow and Harrison, 1997*; *Maxson and Grinstein, 2014*), which makes Phatr3_J27923 a candidate protein involved in acidification of phytotransferrin-rich endosomes in *P. tricornutum*. Notably, V-ATPase transcripts were overrepresented in diatoms following iron enrichment, possibly to account for an increase in the number of endosomes associated with pTF-driven iron acquisition (*Marchetti et al., 2012*), and V-ATPase was shown to be involved in diatom silica deposition vesicle (SDV) acidification (*Hildebrand et al., 2018*; *Yee et al., 2020*). Endosome acidification, perhaps via V-ATPase, may facilitate phytotransferrin-bound ferric, $Fe^{3+}$, iron release, possibly for processing by an iron reducing enzyme similar to the human metalloreductase STEAP3.

Human cellular repressor of E1A-stimulated genes 1 (CREG1) is a glycoprotein involved in embryonic development, growth, differentiation, and senescence as part of the endosomal-lysosomal system (*Ghobrial et al., 2018*; *Schähs et al., 2008*). Contrary to evolutionarily related flavin mononucleotide (FMN)-binding split barrel fold oxidases and oxidoreductases such as pyridoxine 5′-phosphate (PNP) oxidase, an enzyme involved in vitamin $B_6$ metabolism (*Musayev et al., 2003*), CREG1 forms a homodimer unable to bind FMN in vitro suggesting it does not act as an enzyme in vivo (*Sacher et al., 2005*). The lack of FMN in human CREG1 is due to a short loop protruding into the FMN-binding pocket at the dimer interface, and aspartic acid and methionine amino acid residues replacing the corresponding two yeast PNP oxidase (PDB ID: 1CI0) arginines responsible for binding the terminal FMN phosphate (*Sacher et al., 2005*).

CREG1-like protein pTF.CREG1 was proposed to be central to sustained growth of iron-limited *P. tricornutum* cells (*Allen et al., 2008*) or to serve as a cell surface iron receptor (*Lommer et al., 2012*). CREG transcripts were more common in diatom transcriptomes from the Southern Ocean as opposed to non-Southern Ocean regions indicating these proteins are important in coping with iron limitation in this major high-nutrient, low-chlorophyll (HNLC) zone (*Moreno et al., 2018*). This may also explain why more diatom homologs are not present in our phylogenetic tree as transcriptomes from the Marine Microbial Eukaryote Transcriptome Sequencing Project (MMETSP) database used in our phylogenetic analysis were not obtained from iron-limited cultures (*Caron et al., 2017*; *Keeling et al., 2014*). Our protein feature analysis revealed a signal peptide and multiple putative glycosylation sites which suggests that pTF.CREG1 enters the secretory pathway (*Bard and Chia, 2016*). While the nature of pTF.CREG1 recruitment to and colocalization with pTF-rich endosomes in the vicinity of chloroplast periphery—as suggested by our fluorescence microscopy results—is unclear, it is possible that pTF.CREG1-containing vesicles intersect with pTF-rich endosomes en route through the secretory pathway. Vesicular trafficking is highly dynamic and bidirectional (*Naslavsky and Caplan, 2018*; *Progida and Bakke, 2016*), and early endosomal traffic consists of sequential vesicle fusion events (*Brenner, 2012*), so this is at least in principle plausible. Punctate pTF.CREG1 colocalization with pTF suggests that this protein acts in an endosomal-lysosomal system, and this localization pattern is consistent with the observation that proteins with low confidence ASAFind chloroplast prediction (such as pTF.CREG1) can be associated with 'blob-like structures' (BLBs) adjacent to, but not inside, the chloroplast (*Gruber et al., 2015*; *Kilian and Kroth, 2005*). Given that our transgenic colocalization *P. tricornutum* cell lines were neither synchronized nor grown in iron-deplete conditions, it is plausible the protein also acts closer to the cell surface. Likely, but not confirmed, APEX2 orientation away from the cell surface and endosome interior in our proximity labeling experiments, and the reported inability of APEX2-generated phenoxyl radicals to penetrate endomembranes (*Rhee et al., 2013*), would suggest that pTF.CREG1—at some point during its recruitment to pTF-rich endosomes—has to be exposed to the cytosol to promote efficient biotinylation. Alternatively, it is impossible to rule out the possibility that lipid composition of *P. tricornutum* endomembranes is permissive to phenoxyl radical diffusion (*Zulu et al., 2018*).

Domain annotation in pTF.CREG1 (UniProt ID: B7G9B3) encouraged us to perform amino acid sequence and structural alignments with human CREG1 (UniProt ID: O75629, PDB ID: 1XHN). These analyses revealed that pTF.CREG1 lacks at least one arginine amino acid residue required for terminal FMN phosphate binding and likely contains an FMN binding pocket-occluding loop, two key features delineating human CREG1 from the yeast PNP oxidase (*Sacher et al., 2005*). Our hypothesis that pTF.CREG1 is unable to coordinate FMN was initially supported by our protein purifications which were not yellow as expected from a typical flavoprotein (*Theorell, 1935*), and was subsequently strengthened by their catalytic inactivity upon supplementation with biologically relevant flavin cofactors. The demonstrated lack of enzymatic activity in pTF.CREG1 and localization data from in vivo fluorescent reporter experiments represent an important extension of the human CREG1 literature to the single-celled eukaryotic phytoplankton field.

Secreted glycosylated human CREG1 was shown to delay G1/S cell cycle phase transition via direct interaction with mannose 6-phosphate/insulin-like growth factor II receptor (M6P/IGF2R), a receptor internalized via clathrin-mediated endocytosis (*Di Bacco and Gill, 2003*; *Ghosh et al., 2003*; *Han et al., 2009*). This cellular growth inhibition was demonstrated to, at least partially, depend on a tetrapeptide motif within an FMN binding pocket-occluding loop (*Sacher et al., 2005*). In conjunction with our amino acid sequence alignments and structural prediction which point to the conservation of these two features in pTF.CREG1, these studies lend additional support to our data suggesting colocalization and potential direct physical association between phytotransferrin and pTF.CREG1 is non-enzymatic in nature. Internalization of the human transferrin-transferrin receptor complex is not a constitutive, but rather a regulated kinase-activity-dependent process (*Cao et al., 2016*). We currently hypothesize that pTF.CREG1 may serve as a 'checkpoint' to keep phytotransferrin endocytosis at a low level in iron-replete conditions, but when iron is scarce instead positively regulate this pathway. Indeed, transcriptomics experiments show that a second transcriptional peak for *pTF.CREG1* occurs in iron-deplete medium approximately 8 hr prior to the *pTF* peak, a transcriptional dynamic which would be consistent with our hypothesis (*Figure 5—figure supplement 1*; *Smith et al., 2016*). Given that glycosylation in human CREG1 appears to be important for interaction with its cognate receptor (*Han et al., 2011*; *Sacher et al., 2005*; *Schähs et al., 2008*), post-translational modifications, perhaps glycosylation(s), of pTF.CREG1 may further drive this regulatory activity. The presence of Phatr3_J10972, an iron-insensitive pTF.CREG1 paralog, in the *P. tricornutum* genome suggests that CREG-like proteins are implicated in diverse endocytosis pathways in single-celled eukaryotic phytoplankton. However, the association between iron-sensitive pTF.CREG1 and phytotransferrin in *P. tricornutum* implies a specific pTF.CREG1 function in non-reductive iron uptake. It will be very interesting to define the exact pTF.CREG1 role in further cellular and mechanistic studies and compare it to that of its homologs in multicellular organisms. Conceivably, an ancient protein like pTF.CREG1 could have remained at the core of linking cell cycle progression, cellular growth, and nutrient homeostasis across the tree of life.

In contrast to pTF.CREG1, no functional annotation could be assigned to and no structural homology models were generated for pTF.CatCh1 in the Pfam database (*El-Gebali et al., 2019*) and by Phyre2, respectively. This protein has a predicted N-terminal signal peptide and numerous N- and O-glycosylation sites suggesting it enters and traffics along the secretory pathway. The outermost complex plastid membrane in diatoms corresponds to the chloroplast endoplasmic reticulum (cER) and is continuous with the nuclear envelope (*Gibbs, 1981*; *Prihoda et al., 2012*). Nuclear-encoded *P. tricornutum* proteins destined for the periplastidial compartment (vastly reduced endosymbiont cytoplasm between the inner two and the outer two complex plastid membranes) are first cotranslationally imported into the cER lumen (*Grosche et al., 2014*; *Peschke et al., 2013*). Our microscopy results show that pTF.CatCh1 is localized on the chloroplast margin, perhaps anchored to the outer chloroplast (cER) membrane with its C-terminal transmembrane domain, thus raising a possibility that pTF.CatCh1 gets to its final destination via cotranslational membrane insertion.

The cysteine-rich domain contains one CXC and one CXXC motifs which are nearly invariably present in pTF.CatCh1 homologs. Two cysteines spaced by two non-cysteine amino acids (the CXXC motif) are commonly found in metal-binding proteins, including those that bind iron (*Fomenko et al., 2008*; *Gladyshev et al., 2004*). This motif in pTF.CatCh1 is preceded by an aspartic acid residue resulting in DCXXC which is known to be particularly permissive to binding divalent metal ions (*Figure 7—figure supplement 1A*; *Banci et al., 2002*; *Banci et al., 2006*).

Histidine, tyrosine, methionine, aspartic acid, and glutamic acid residues can also be involved in iron cation coordination, but none of these are conserved across pTF.CatCh1 homologs (*Figure 7—source data 1*; *Dokmanić et al., 2008*). Given the nearly completely conserved C(X)XC motifs and additional six 100% conserved cysteine residues, it is conceivable they are critical to pTF.CatCh1 activity.

Based on the fluorescence microscopy data, the conserved cysteine amino acid residues and cysteine-containing motifs, and given that disordered protein domains—two are predicted to flank the pTF.CatCh1 transmembrane domain—are mediators of protein-protein interactions and play an important role in the formation of protein complexes (*Uversky, 2015b*; *Uversky, 2016*), we are considering two hypotheses for in vivo pTF.CatCh1 function.

First, pTF.CatCh1 may serve as a metallochaperone that binds ferrous, $Fe^{2+}$, iron downstream of the endosomal ferric, $Fe^{3+}$, iron reduction step, and directs it to the chloroplast interior. Metallochaperones are intracellular proteins responsible for protecting and guiding metal ions on their way to correct metalloenzyme sinks (*O'Halloran and Culotta, 2000*). They engage in protein-protein interactions and help prevent metal-induced toxicity caused by free radical-generating Fenton reactions which are especially common with iron and copper (*Philpott and Jadhav, 2019*; *Rosenzweig, 2002*; *Valko et al., 2005*).

Second, pTF.CatCh1 may recruit pTF to the chloroplast periphery via one of its two C-terminal disordered regions flanking the predicted transmembrane domain, and thus serve as a docking site for iron-laden pTF. It could achieve this via the 'fly-casting mechanism' which involves binding-induced protein folding (*Shoemaker et al., 2000*). The observed colocalization of pTF.CatCh1 with pTF may indicate that $Fe^{3+}$ can be offloaded directly from pTF to pTF.CatCh1 which would imply an additional, non-endosomal, intracellular $Fe^{3+}$ reduction step. Notably, a chloroplast-associated $Fe^{3+}$ reduction pathway mediated by $Fe^{3+}$ chelate reductase FRO7 is present in *Arabidopsis thaliana* (*Jeong et al., 2008*). Abberant cellular localization of heterologously expressed fluorescent protein fusions is a widely observed and acknowledged phenomenon (*Jensen, 2012*), but we note that pTF localization on the chloroplast margin insdistinguishable from that in this study was reported by *McQuaid et al., 2018* suggesting it is biologically relevant. Similarly to *pTF.CREG1*, *pTF.CatCh1* exhibits a second transcriptional peak in iron-deplete medium approx. 16 hr prior to the *pTF* peak (*Figure 5—figure supplement 1*). It would make sense to synthesize a protein critical for iron-laden pTF recruitment to the chloroplast margin prior to significant increase in basal pTF endocytosis.

All in all, our data suggest pTF.CatCh1 is a chloroplast margin-localized, pTF-associated, *P. tricornutum* protein with a possible role in chloroplast iron homeostasis. The two hypotheses are put forth here to help frame its further genetic, cellular, biochemical, and structural dissection.

Chloroplasts are the most iron-rich system in plant cells (*López-Millán et al., 2016*). Iron—in particular in the form of iron–sulfur (Fe–S) clusters—serves as an essential cofactor for the photosynthetic electron transfer chain, chlorophyll biosynthesis, and chloroplast protein import (*Balk and Schaedler, 2014*; *Marchand et al., 2018*; *Soll and Schleiff, 2004*). A complex vesicle-based system possibly in perpetual dynamic exchange with cytosolic vesicles is present in chloroplasts (*Hertle et al., 2020*; *Lindquist and Aronsson, 2018*), an endoplasmic reticulum to Golgi to chloroplast protein trafficking pathway exists (*Radhamony and Theg, 2006*; *Villarejo et al., 2005*), and additional hypotheses for such specialized vesicle-mediated plastid targeting of proteins are established (*Baslam et al., 2016*). Therefore, a direct link between endocytic internalization of phytotransferrin-bound iron and its offloading to or adjacent to chloroplasts as suggested by our results seems plausible. 'Kiss and run' mechanism that allows erythroid cell mitochondria to access iron via direct interaction with transferrin-rich endosomes further underscores this idea (*Hamdi et al., 2016*). Analysis of our mass spectrometry (MS) hits that were not co-expressed with pTF-mCherry reveals three additional proteins—ISIP1 (Phatr3_J55031), FBP1 (Phatr3_J46929), and FBAC5 (Phatr3_J41423)—that support the idea of a direct cell surface-to-chloroplast iron trafficking pathway in *Phaeodactylum tricornutum*.

*ISIP1* (iron starvation induced protein 1; Phatr3_J55031) gene expression in *P. tricornutum* and other marine microeukaryotes is induced by iron limitation (*Allen et al., 2008*; *Kazamia et al., 2018*; *Marchetti et al., 2012*). *P. tricornutum* ISIP1—the second most enriched protein in our MS dataset—has a role in siderophore-bound iron uptake and colocalizes with phytotransferrin (pTF) adjacently to the chloroplast in an 'iron-processing compartment' (*Kazamia et al., 2019*; *Kazamia et al., 2018*). FBP1 (Phatr3_J46929)—iron-sensitive siderophore ferrichrome-binding protein with punctate

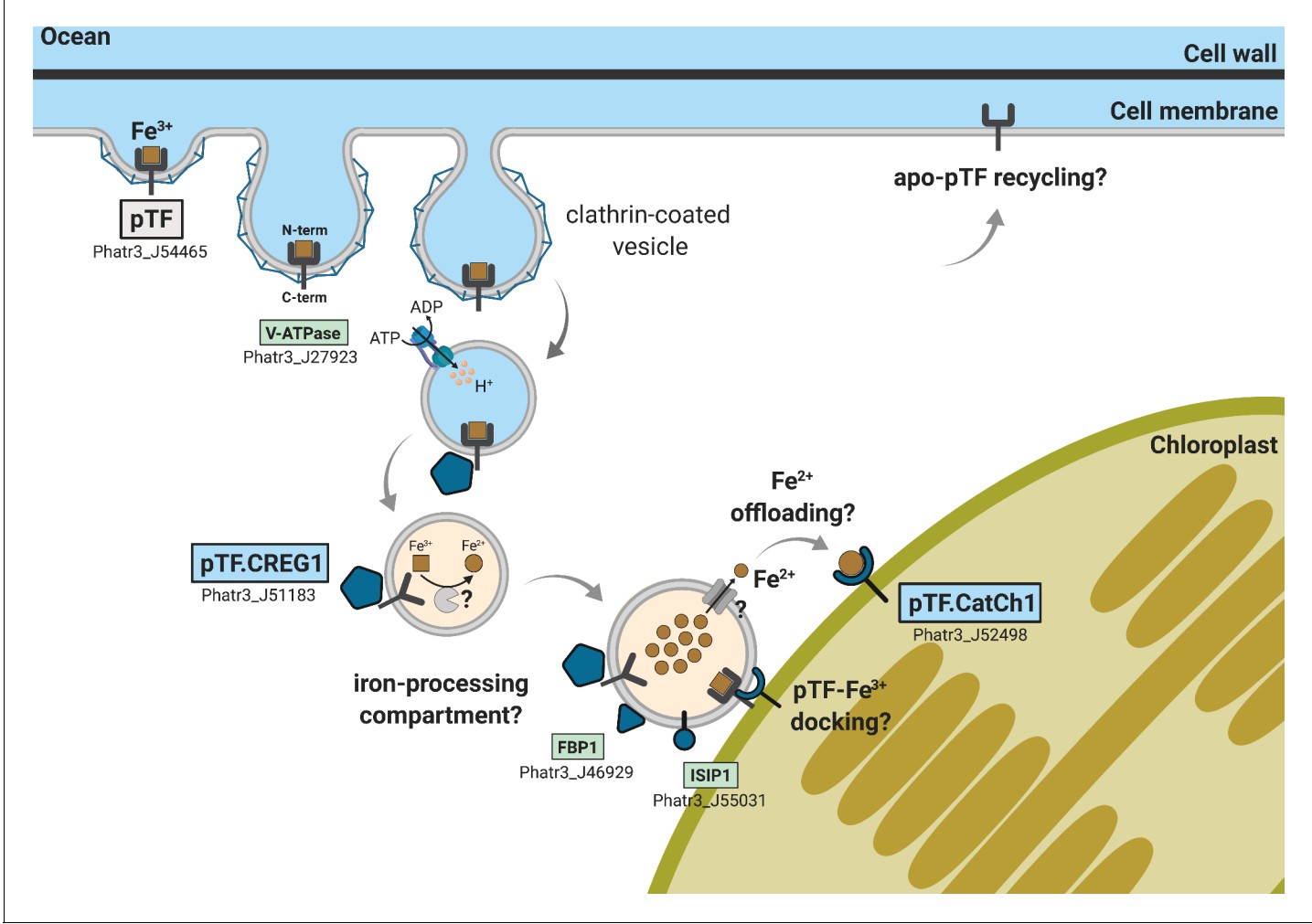

**Figure 8.** Emerging view of intracellular inorganic iron allocation pathway in *Phaeodactylum tricornutum.* We propose a model for clathrin-mediated internalization of phytotransferrin (pTF)-bound ferric, $Fe^{3+}$, iron, and its trafficking to the chloroplast periphery. In this model, pTF releases $Fe^{3+}$ after undergoing a structural change upon endosome acidification driven by a vacuolar-type $H^+$-ATPase. pTF.CREG1 may regulate this process at the endosome periphery and perhaps also in earlier endocytosis stages closer to the cell surface. It is unclear how intravesicular reduction of ferric iron is achieved and how the resulting ferrous iron gets exported. Given that no DMT1/NRAMP2 homologs are present in *P. tricornutum*, Zrt- and Irt-like proteins (ZIPs)—known to export $Fe^{2+}$ from intracellular compartments—may mediate the latter (***Blaby-Haas and Merchant, 2012***; ***Kustka et al., 2007***; ***Lampe et al., 2018***). Ferrous iron could be offloaded to pTF.CatCh1 on the outer chloroplast membrane and ultimately transported further into the chloroplast interior for cellular assimilation and use (e.g. iron–sulfur cluster biosynthesis for incorporation into photosystems I and II). pTF.CatCh1 may alternatively (or additionally) be responsible for pTF recruitment to the chloroplast margin. The enrichment of FBP1 and ISIP1—both previously shown to localize to intracellular vesicles in *P. tricornutum* (***Coale et al., 2019***; ***Kazamia et al., 2019***)—in our proximity proteomics experiment suggests the whole pathway intersects with that for internalizing organic, siderophore-bound, iron. Gray panel: pytotransferrin (pTF). Blue panels: proteins colocalized with pTF. Green panels: additional proteins enriched in our APEX2 experiment. One phytotransferrin-$Fe^{3+}$ complex is shown for clarity. Orange vesicle interior indicates pH drop due to ATPase activity. Detailed chloroplast membrane layers are omitted (diatom chloroplasts contain four membranes). Created with BioRender.com.

The online version of this article includes the following figure supplement(s) for figure 8:

**Figure supplement 1.** Outstanding biological problems in established model marine diatoms poised to be advanced with proximity-dependent proteomic mapping approaches.

intracellular localization and central to the carbonate-independent iron acquisition pathway in *P. tricornutum* (*Allen et al., 2008*; *Coale et al., 2019*)—was another highly enriched protein in our dataset indicating the internalization pathways for inorganic (i.e. phytotransferrin-bound) and organic (i.e. siderophore-bound) iron in *P. tricornutum* may be physically coupled intracellularly and directed to iron-accumulating vesicles close to the chloroplast. Metal homeostasis by such discrete membranous compartments is widespread in nature (*Blaby-Haas and Merchant, 2014*). Copper- and zinc-storing compartments—cuprosomes and zincosomes, respectively—are present in the model chlorophyte microalga *Chlamydomonas reinhardtii* (*Aron et al., 2015*; *Hong-Hermesdorf et al., 2014*; *Merchant, 2019*), and lipid-bound iron-accumulating ferrosome compartments exist in diverse bacteria and archaea (*Grant and Komeili, 2020*).

Finally, the most enriched protein in our MS dataset was FBAC5 (Phatr3_J41423)—iron-sensitive iron-independent class I fructose-biophosphate aldolase—which was shown to be localized to the *P. tricornutum* pyrenoid—proteinaceous RuBisCO-containing subchloroplastic organelle—and proposed to link Calvin-Benson-Bassham cycle activity with the $CO_2$ concentrating mechanism (*Allen et al., 2012*; *Matsuda et al., 2017*). FBAC5 is one of the most reliable markers for diatom iron stress as demonstrated by laboratory culture (*Cohen et al., 2018*; *Lommer et al., 2012*), microcosm incubation (*Cohen et al., 2017a*; *Cohen et al., 2017b*), and field studies (*Bertrand et al., 2015*); however, its exact role has remained elusive. Fructose-biphosphate aldolases (FBAs) are glycolytic enzymes known for their ability to moonlight (*Boukouris et al., 2016*), a term for proteins with multiple physiologically relevant functions encoded by one polypeptide chain (*Jeffery, 2018*). Moonlighting FBAs have roles in actin cytoskeleton dynamic, cellular growth arrest, and apoptosis (in human cancer cell lines; *Gizak et al., 2019*), transcriptional regulation (in yeast; *Cieśla et al., 2014*), and endosome acidification via direct interaction with vacuolar-type ATPase (in a mouse cell line; *Merkulova et al., 2011*). These reported non-catalytic FBA activities suggest that FBAC5 acting outside of the *P. tricornutum* chloroplast, perhaps in the endosomal-lysosomal system as our data would imply, is possible.

Our initial characterization of pTF.CREG1 and pTF.CatCh1, together with the discussion on the four other mass spectrometry hits, enable proposition of a model connecting cell surface ferric iron binding, internalization, and intracellular trafficking in *Phaeodactylum tricornutum* with many outstanding questions ripe for future investigation (*Figure 8*). Importantly, it remains to be determined which protein is responsible for intracellular ferric, $Fe^{3+}$, iron reduction in *P. tricornutum*, activity that would be consistent with the non-reductive cell surface ferric, $Fe^{3+}$, iron uptake proposed for phytotransferrin (pTF) (*McQuaid et al., 2018*; *Morrissey et al., 2015*). Perhaps notably, neither of the five proteins with clear ferric reductase annotation (FRE1–5, Phatr3 IDs: J54486, J46928, J54940, J54409, J54982) nor additional five hypothetical ferric reductases (*Coale et al., 2019*; *Smith et al., 2016*) were present in our mass spectrometry data. Further proximity-dependent proteomic mapping experiments will be essential for determining the full set of proteins in the non-reductive iron acquisition pathway in *P. tricornutum* as enzymatic $Fe^{3+}$ reduction may happen later in the pathway and/or in a separate subcellular compartment.

In conclusion, the identified proteins and the associated model we introduce in this work represent an important advance for the marine phytoplankton field, given that diatoms are central to the study of cellular iron homeostasis and its link to fluctuating iron levels in the ocean. The presented insights are vital as our understanding of intracellular iron trafficking is relatively incomplete even in significantly more established model systems such as mammalian cells, and our knowledge of chloroplast iron uptake is in its infancy (*Blaby-Haas and Merchant, 2012*; *Philpott and Jadhav, 2019*). We anticipate further molecular dissection of our proteomics dataset will not only provide details on the transformation dynamic of acquired ferric iron and its cellular sinks but will thereby also illuminate the extent to which phytotransferrin pathway mirrors the metazoan transferrin cycle.

Implementing the use of APEX2 in a model marine diatom is itself a significant and timely advance as the field is expanding and moving from broad -omics-based surveys to detailed molecular studies (*Faktorová et al., 2020*; *Falciatore et al., 2020*). APEX2 and related molecular tools hold great promise to dissect additional key subcellular compartments in diatoms such as pyrenoids and silica deposition vesicles (*Figure 8—figure supplement 1*; *Barrett et al., 2021*; *Hildebrand et al., 2018*; *Matsuda et al., 2017*; *Wang and Jonikas, 2020*), two exciting future avenues for these

single-celled heterokonts uniquely positioned at the nexus of evolutionary cell biology and blue biotechnology.

# Materials and methods

## Key resources table

| Reagent type (species) or resource | Designation | Source or reference | Identifiers | Additional information |
|---|---|---|---|---|
| Gene (*Phaeodactylum tricornutum*) | *pTF; ISIP2a* | PMID:29539640 | Ensembl-Protists: Phatr3_J54465 | Jeffrey B. McQuaid |
| Gene (*Phaeodactylum tricornutum*) | *pTF.CREG1* | This paper | Ensembl-Protists: Phatr3_J51183 | The first subcellular localization and enzymatic activity reports for this *P. tricornutum* protein |
| Gene (*Phaeodactylum tricornutum*) | *pTF.CatCh1* | This paper | Ensembl-Protists: Phatr3_J52498 | The first subcellular localization report for this *P. tricornutum* protein |
| Gene (*Phaeodactylum tricornutum*) | *pTF.ap1* | This paper | Ensembl-Protists: Phatr3_J54986 | The first subcellular localization report for this *P. tricornutum* protein |
| Gene (*Glycine max*) | *APEX2* | PMID:25419960 | NCBI-Gene:553156 | APEX2 mutations relative to wild-type APX (encoded by NCBI-Gene:553156): K14D, W41F, E112K, A134P |
| Strain, strain background (*Phaeodactylum tricornutum*) | *Phaeodactylum tricornutum; P. tricornutum*; WT *P. tricornutum* | NCMA | Catalog #:CCMP632 Starter Culture 2 × 15 ml | CCMP632 is synonymous with CCMP2561 and CCAP 1055/1 |
| Strain, strain background (*Escherichia coli*) | EPI300 | Lucigen | Catalog #:EC300110 | Electrocompetent; recommended for bacterial conjugation of diatoms |
| Strain, strain background (*Escherichia coli*) | BL21 | NEB | Catalog #:C2530H | Chemically competent |
| Genetic reagent (*Phaeodactylum tricornutum*) | Δ*pTF P. tricornutum* | PMID:29539640 | | Jeffrey B. McQuaid; TALEN-generated pTF knockout *P. tricornutum* strain; available from the Allen Lab |
| Transfected construct (*Phaeodactylum tricornutum*) | pJT_NR_pTF-AP2 | This paper | | See *Figure 2B*, Materials and methods, and *Supplementary file 1*—Table S4; available from the Allen Lab |
| Transfected construct (Δ*pTF Phaeodactylum tricornutum*) | pJT_native_pTF-mCherry | This paper | | See Materials and methods and *Supplementary file 1*—Table S4; available from the Allen Lab |
| Transfected construct (*Phaeodactylum tricornutum*) | pJT_pTF-mCherry_pTF.CREG1-EYFP | This paper | | See Materials and methods and *Supplementary file 1*—Table S4; available from the Allen Lab |

*Continued on next page*

*Continued*

| Reagent type (species) or resource | Designation | Source or reference | Identifiers | Additional information |
|---|---|---|---|---|
| Transfected construct (*Phaeodactylum tricornutum*) | pJT_pTF-mCherry _pTF.CatCh1-EYFP | This paper | | See Materials and methods and *Supplementary file 1*—Table S4; available from the Allen Lab |
| Transfected construct (*Phaeodactylum tricornutum*) | pJT_pTF-mCherry _pTF.ap1-EYFP | This paper | | See Materials and methods and *Supplementary file 1*—Table S4; available from the Allen Lab |
| Transfected construct (*Escherichia coli* BL21) | pJT_Δ31_pTF.CREG1-His6 | This paper | | See Materials and methods and *Supplementary file 1*—Table S4; available from the Allen Lab |
| Antibody | Anti-pTF (rabbit monoclonal) | PMID:29539640 | | WB (1:10,000 of 1.14 mg/mL stock); custom-made at OriGene; available from the Allen Lab |
| Recombinant DNA reagent | pPtPBR1 | Addgene | Catalog #:80388 | Episomal vector for bacterial conjugation of *P. tricornutum*; available from the Allen Lab |
| Recombinant DNA reagent | pTA-Mob | PMID:24595202 | | Mobilization plasmid for bacterial conjugation of diatoms; available from the Allen Lab |
| Recombinant DNA reagent | PtpBAD-CTHF | PMID:30262498 | | *Escherichia coli* protein expression vector; available from the Allen Lab |
| Sequence-based reagent | JT01 | This paper | PCR primer | CGAATCAGGATCTAAAATGAACGCACGTCTGCGACCTGAGCAA; see Materials and methods and *Supplementary file 1*—Table S3 |
| Sequence-based reagent | JT02 | This paper | PCR primer | GTCGCTTCACGTTCGCTC; see Materials and methods and *Supplementary file 1*—Table S3 |
| Sequence-based reagent | JT03 | This paper | PCR primer | GATACGCGAGCGAACGTGAAGCGACTCACGTAGTGAAGTGATGTTG; see Materials and methods and *Supplementary file 1*—Table S3 |
| Sequence-based reagent | JT04 | This paper | PCR primer | TTCCAGACGTAGAACCACTCCCTTTGATAGGAGTGCTGCCAGTG; see Materials and methods and *Supplementary file 1*—Table S3 |

*Continued on next page*

*Continued*

| Reagent type (species) or resource | Designation | Source or reference | Identifiers | Additional information |
|---|---|---|---|---|
| Peptide, recombinant protein | PrimeSTAR GXL DNA Polymerase | Takara Bio | Catalog #:R050B | Recommended for amplifying (parts of) diatom episomal vectors |
| Peptide, recombinant protein | Bovine Serum Albumin, Biotinylated | Thermo Fisher Scientific | Catalog #:29130 | |
| Peptide, recombinant protein | Streptavidin, horseradish peroxidase (HRP) conjugate | Thermo Fisher Scientific | Catalog #:S911 | WB (1:15,000) |
| Peptide, recombinant protein | WesternSure Pre-stained Chemiluminescent Protein Ladder | Li-COR | Catalog #:926–98000 | Recommended for streptavidin blotting |
| Commercial assay or kit | Phire Plant Direct PCR Master Mix | Thermo Fisher Scientific | Catalog #:F160L | Recommended for diatom genotyping |
| Commercial assay or kit | Phire Plant Direct PCR Kit | Thermo Fisher Scientific | Catalog #:F130WH | Recommended for diatom genotyping |
| Commercial assay or kit | WesternBreeze Chemiluminescent Kit, anti-rabbit | Thermo Fisher Scientific | Catalog #:WB7106 | |
| Commercial assay or kit | TMT10plex Isobaric Label Reagent Set | Thermo Fisher Scientific | Catalog #:90406 | |
| Chemical compound, drug | Amplex UltraRed Reagent | Thermo Fisher Scientific | Catalog #:A36006 | |
| Chemical compound, drug | 3,3'-Diaminobenzidine (DAB) | Sigma-Aldrich | Catalog #:D8001 | |
| Chemical compound, drug | D-(+)-Biotin-tyramine amide (biotin-phenol) | Berry and Associates | Catalog #:BT 1015 | |
| Software, algorithm | SEQUEST (v. 28, rev. 12) algorithm | PMID:24226387 | | Algorithm for matching tandem mass spectra with peptide sequences |
| Software, algorithm | JMP | SAS Institute | | Interactive statistical discovery software |
| Software, algorithm | R | R Core Team | | Free software environment for statistical computing and graphics |
| Software, algorithm | ASAFind | PMID:25438865 | | Plastidial protein localization prediction tool for algae with red secondary plastids; available at https://rocaplab.ocean.washington.edu/tools/asafind/ |
| Software, algorithm | BLASTP | PMID:2231712 | | Algorithm for identifying homologous proteins |
| Software, algorithm | HMMER | PMID:9918945 | | Software for identifying homologous proteins (or nucleotide sequences) using profile hidden Markov models |

*Continued on next page*

*Continued*

| Reagent type (species) or resource | Designation | Source or reference | Identifiers | Additional information |
|---|---|---|---|---|
| Software, algorithm | MAFFT | PMID:12136088 | | Multiple amino acid sequence alignment algorithm based on fast Fourier transform |
| Software, algorithm | SeaView 4 | PMID:19854763 | | Graphical User Interface (GUI)-based molecular phylogeny software |
| Software, algorithm | IQ-TREE | PMID:25371430 | | Algorithm for inferring phylogenetic trees by maximum likelihood |
| Software, algorithm | Phyre2 | PMID:25950237 | | Web server to predict and analyze protein structure, function and mutations; available at http://www.sbg.bio.ic.ac.uk/~phyre2/html/page.cgi?id=index |
| Software, algorithm | UCSF Chimera version 1.11.1 | PMID:15264254 | | Software for interactive visualization and analysis of molecular structures and related data |
| Other | Yeast Vacuole Membrane Marker MDY-64 | Thermo Fisher Scientific | Catalog #:Y7536 | Membrane stain |
| Other | Pierce Streptavidin Magnetic Beads | Thermo Fisher Scientific | Catalog #:88816 | Protein-coated iron oxide microparticles |
| Other | *Phaeodactylum tricornutum* proteome | UniProt | Proteome-ID: UP000000759 | Reference proteomic database |
| Other | Phatr3 *P. tricornutum* genomic database | Ensembl Protists | Genome-Assembly: ASM15095v2 | Reference genomic database |
| Other | Transcriptomic data | PMID:27973599 | S1 Dataset. Active transcriptome and assignment of genes to WGCNA modules and response types. | *P. tricornutum* transcriptomic dataset; see *Figure 5—figure supplement 1—source data 1* |
| Other | Marine Microbial Eukaryote Transcriptome Sequencing Project (MMETSP) database | PMID:24959919 | NCBI-BioProject: PRJNA231566 | Transcriptomic database |

## Key resources table

The quantitative mass spectrometry proteomics data used to generate *Figure 4—source data 1* and *Figure 4—figure supplement 1—source data 1* have been deposited to the ProteomeXchange Consortium via the PRIDE partner repository (*Perez-Riverol et al., 2019*) with the dataset identifier PXD018022.

The amino acid sequence alignment files *Figure 6—source data 1*, *Figure 7—source data 1*, and *Figure 7—figure supplement 2—source data 1* can be read and interrogated with Alignment-Viewer available at https://alignmentviewer.org/.

## Vector cloning
### pJT_NR_pTF-AP2

Gibson Assembly (*Gibson et al., 2009*) was performed with three DNA fragments: (1) linearized pPtPBR1 episome backbone (Addgene, Cambridge, MA, plasmid pPtPBR1, Catalog #80388) opened ~280 bp downstream of the tetracycline resistance gene, (2) amplicon with nitrate reductase

gene (*NR*) promoter, 5′ *NR* untranslated region (UTR), and *pTF*, and (3) gBlocks Gene Fragment (Integrated DNA Technologies (IDT), Coralville, IA) with linker sequence, codon optimized *APEX2* (using IDT Codon Optimization Tool and selecting '*Thalassiosira pseudonana*' from the 'Organism' drop-down menu), 3′ *NR* UTR, and *NR* terminator.

### pJT_native_pTF-mCherry

Gibson Assembly was performed with three DNA amplicons: (1) pPtPBR1 episome backbone split into two fragments and (2) expression cassette including *pTF*, *mCherry*, and *pTF* promoter and terminator.

### pJT_pTF-mCherry_MS hit-EYFP

pJT_native_pTF-mCherry was split into two fragments via PCR stitching (keeping ampicillin resistance gene split). Genes corresponding to MS hits were amplified using cDNA from iron-starved WT *P. tricornutum* cells and assembled with *EYFP* and *Phatr3_J23658* (flavodoxin-encoding gene) promoter and terminator through two PCR stitching rounds into a single fragment. Gibson Assembly was then performed to combine all three final amplicons. Detailed assembly scheme is presented in *Figure 5—figure supplement 2A*.

### pJT_Δ31_pTF.CREG1-His6

Cloning of gene fragments into the *E. coli* protein expression vector PtpBAD-CTHF was performed as described previously (*Brunson et al., 2018*). Briefly, PtpBAD-CTHF was linearized by digestion with XhoI (New England Biolabs (NEB), Ipswich, MA, Catalog #R0146S) and the resulting DNA was column purified. *Δ18_pTF.CREG1* gene was obtained by PCR from *P. tricornutum* gDNA with PrimeSTAR GXL DNA Polymerase (Takara Bio, Kusatsu, Japan, Catalog #R050B) and primer set JT31/JT32 to incorporate the appropriate Gibson Assembly overhangs and remove the predicted N-terminal signal peptide (amino acid residues 1–18). Insertion of truncated *pTF.CREG1* into linearized PtpBAD-CTHF was performed using Gibson Assembly Master Mix (NEB, Catalog #E2611S), 1 µL of the Gibson Assembly reaction mixture was transformed via heat shock into chemically competent NEB 5-alpha cells (NEB, Catalog #C2988J), and cells were incubated on lysogeny broth with 10 µg/mL tetracycline (LB-Tet10) 1% agar plates overnight at 37°C. Transformants were screened by colony PCR using the primer set JT37/JT38 and Sapphire polymerase (Takara Bio), and positive clones were selected for outgrowth. Isolated plasmids were sequence-validated by Sanger sequencing (Eurofins, Luxembourg, Luxembourg). A sequence-validated clone was designated as PtpBAD-Δ18_pTF.CREG1-CTHF and transformed into chemically competent BL21 *E. coli* cells (NEB, Catalog #C2530H) which were spread on LB-Tet10 1% agar plates. The resulting transformants were used for subsequent Δ18_pTF.CREG1 expression experiments. Following unsuccessful expression testing of the Δ18 N-terminal truncation construct, an additional set of genes encoding pTF.CREG1 with N-terminal truncations (Δ26, Δ31, Δ39, Δ43) was generated by PCR using PrimeStar GXL DNA Polymerase (Takara Bio), primer sets JT33/JT32, JT34/JT32, JT35/JT32, JT36/JT32, and PtpBAD-Δ18_pTF.CREG1-CTHF as a template. Assembly, colony PCR screening, plasmid isolation, sequencing, and transformation into chemically competent BL21 *E. coli* cells of all additional expression vectors, including PtpBAD-Δ31_pTF.CREG1-CTHF (=pJT_Δ31_pTF.CREG1-His6), was performed as above. An additional construct encoding pTF.CREG1 with an N-terminal His6 and Δ39 N-terminal truncation was built (PtpBAD-NTH-Δ39_pTF.CREG1) using the vector NTH-PtpBAD (constructed similarly to PtpBAD-CTHF; see *Brunson et al., 2018* and *Savitsky et al., 2010*), but was not pursued beyond initial expression testing.

pTF (*Phatr3_J54465*) in all episomes was in its native form (3 exons, 2 introns). Molecular cloning primers are listed in *Supplementary file 1*—Table S3. Vector details, further amplicon information, and fusion protein sequences are catalogued in *Supplementary file 1*—Table S4.

### Related resource

Turnšek J, Gholami P. 2017. Guidelines for highly efficient construction of diatom episomes using Gibson Assembly. *protocols.io*. DOI: dx.doi.org/10.17504/protocols.io.jy7cpzn.

### Diatom culturing, conjugation, and genotyping

#### Culturing

Sequenced *Phaeodactylum tricornutum* strain CCMP632 (synonymous to CCMP2561 and CCAP 1055/1; National Center for Marine Algae and Microbiota (NCMA), East Boothbay, ME) was used throughout the study and grown at 18°C, 300 µmol quanta m$^{-2}$ s$^{-1}$, and a 10 hr:14 hr dark:light cycle in biotin-free L1 medium prepared by mixing 1 L Aquil salts, 2 mL nitrate and phosphate (NP) nutrient stock, 1 mL trace metal stock, and 1 mL thiamine hydrochloride and cyanocobalamin (TC) stock unless otherwise noted. Preparation of Aquil salts: 0.5 L anhydrous salts (0.5 L Milli-Q, 24.5 g NaCl, 4.09 g $Na_2SO_4$, 0.7 g KCl, 0.2 g $NaHCO_3$, 0.1 g KBr, 900 µL 33.3 mg/mL $H_3BO_3$ stock, 300 µL 10 mg/mL NaF stock) and 0.5 L hydrous salts (0.5 L Milli-Q, 11.1 g $MgCl_2 \times 6H_2O$, 1.54 g $CaCl_2 \times 2H_2O$, 100 µL 170 mg/mL $SrCl_2 \times 6H_2O$ stock) were prepared separately, combined, filter sterilized (0.2 µm), and stored at room temperature. Preparation of NP nutrient stock: 37.5 g $NaNO_3$ and 2.5 g $NaH_2PO_4$ were dissolved in 100 mL Milli-Q, filter sterilized (0.2 µm), and stored at 4°C. Preparation of trace metal stock (for 1 L 1000x stock): 3.15 g $FeCl_3 \times 6H_2O$, 4.36 g $Na_2EDTA \times 2H_2O$, 0.25 mL 9.8 g/L $CuSO_4 \times 5H_2O$, 3.0 mL 6.3 g/L $Na_2MoO_4 \times 2H_2O$, 1.0 mL 22 g/L $ZnSO_4 \times 7H_2O$, 1.0 mL 10 g/L $CoCl_2 \times 6H_2O$, 1.0 mL 180 g/L $MnCl_2 \times 4H_2O$, 1.0 mL 1.3 g/L $H_2SeO_3$, 1.0 mL 2.7 g/L $NiSO_4 \times 6H_2O$, 1.0 mL 1.84 g/L $Na_3VO_4$, 1.0 mL 1.94 g/L $K_2CrO_4$, and Milli-Q up to 1 L were combined, filter sterilized (0.2 µm), and kept at 4°C. Preparation of TC stock: 20 mg thiamine hydrochloride and 0.1 mL 1 g/L cyanocobalamin stock were mixed in 100 mL Milli-Q. The resulting solution was stored at 4°C. Δ*pTF P. tricornutum* cell line (maintained in the Allen Lab; *McQuaid et al., 2018*) was additionally supplemented with 200 µg/mL nourseothricin (GoldBio, Saint Louis, MO, Catalog #N-500–1). All transconjugant *P. tricornutum* cell lines were supplemented with 50 or 100 µg/mL phleomycin (InvivoGen, San Diego, CA, Catalog #ant-ph-10p).

#### Conjugation

(1) Bacterial donor preparation: Chemically competent pTA-Mob-containing TransforMax EPI300 *E. coli* cells (*Strand et al., 2014*; Lucigen, Middleton, WI, Catalog #EC300110) were transformed via heat shock with sequence-verified pPtPBR1 episomes. Transformants were selected on gentamycin-, carbenicillin-, and tetracycline-containing LB 1% agar plates. 3 mL overnight LB cultures supplemented with antibiotics were inoculated from glycerol stocks. (2) *P. tricornutum* preparation: ~2×10$^8$ *P. tricornutum* cells (in 200 µL) in mid- to late-exponential phase were spread on pre-dried (i.e. plates with lid half open and kept in the laminar flow hood for at least 90 min) ½ L1 1% agar plates with 5% LB and left growing for 1 or 2 days. Plates were additionally supplemented with 200 µg/mL nourseothricin for conjugation of Δ*pTF P. tricornutum* cells. (3) Conjugation: Overnight donor bacterial cultures were diluted 1:50 in 25 mL LB supplemented with antibiotics, grown at 37°C until OD$_{600}$ 0.8–1, spun down, resuspended in 150 µL Super Optimal broth with Catabolite repression (SOC) medium, and spread as evenly as possible on top of a *P. tricornutum* lawn. Plates with donor-*P. tricornutum* co-culture were first left in dark and 30°C for 90 min, then for 1 or 2 days at standard growth conditions. (4) Selection: Co-culture lawn was scraped off of plates with 1 mL fresh L1 medium, transferred to a microcentrifuge tube, and 200 µL spread on pre-dried (see above) ½ L1 1% agar plates with 50 or 100 µg/mL phleomycin. Plates were additionally supplemented with 200 µg/mL nourseothricin for conjugation of Δ*pTF P. tricornutum* cells. Porous adhesive tape was used to seal the plates and transconjugants emerged after ~10 days of incubation under standard growth conditions. Please see *Karas et al., 2015* and *Diner et al., 2016* for further description of diatom conjugation.

#### Genotyping

Candidate transconjugant colonies were inoculated in 300 µL L1 medium supplemented with 50 or 100 µg/mL phleomycin (and 200 µg/mL nourseothricin in case of Δ*pTF P. tricornutum* cells) and typically grown for ~1 week. 0.5 µL liquid culture was then genotyped using either Phire Plant Direct PCR Master Mix (Thermo Fischer Scientific, Waltham, MA, Catalog #F160L) or Phire Plant Direct PCR Kit (Thermo Fischer Scientific, Catalog #F130WH). 200 µL of each genotype-positive cell line was passaged in 30 mL L1 medium supplemented with 50 or 100 µg/mL phleomycin (and 200 µg/mL nourseothricin in case of Δ*pTF P. tricornutum* cells).

Related resource

Turnšek J. 2017. Simple and rapid genotyping of marine microeukaryotes. *protocols.io*. DOI: dx.doi.org/10.17504/protocols.io.jcdcis6.

## RNA extraction and cDNA synthesis

### Diatom culture

100 mL WT *P. tricornutum* culture was grown in iron-deplete conditions (L1 medium with 7.5 nM total iron) for two weeks. Cells were then centrifuged, supernatants discarded, pellets flash frozen in liquid nitrogen, and stored at −80°C.

### RNA extraction

Direct-zol RNA Miniprep Plus RNA extraction kit was used (Zymo Research, Irvine, CA, Catalog #R2070). Briefly, cell pellets were resuspended in 800 μL Trizol, equal amount of 100% ethanol, and centrifuged in spin columns. Columns were then washed with 400 μL RNA Wash Buffer followed by on-column DNA digestion with 5 μL DNase I in 75 μL DNA Digestion Buffer for 15 min at RT, washing twice with 400 μL Direct-zol RNA PreWash, and once with 700 μL RNA Wash Buffer. RNA was eluted with 50 μL DNase/RNase-Free Water, RNA integrity number (RIN) evaluated with 2200 TapeStation (Agilent Technologies, Santa Clara, CA; measured RIN was 7.0), concentration estimated with Qubit 2.0 Fluorometer (Thermo Fischer Scientific; measured concentration was 27.6 ng/μL), and samples stored at −80°C.

### cDNA synthesis

cDNA synthesis kit from Thermo Fischer Scientific was used (SuperScript III First-Strand Synthesis System, Catalog #18080–051). Briefly, 1 μL total RNA was combined with 0.5 μL Oligo(dT)$_{20}$ Primer, 0.5 μL 10 mM dNTP Mix, and 3 μL nuclease-free water followed by 5 min and 1 min incubation at 65°C and on ice, respectively. After 5 μL cDNA Synthesis Mix (1 μL 10x Reverse Transcription buffer, 2 μL 25 mM MgCl$_2$, 1 μL 0.1 M DTT, 0.5 μL RNase OUT, 0.5 μL SuperScript III Reverse Transcriptase) was added, samples were incubated for 50 min and 5 min at 50°C and 85°C, respectively, then chilled on ice. Finally, 1 μL RNase H was added and sample kept for 20 min at 37°C. cDNA was stored at −20°C until use.

## MDY-64 labeling and imaging

Ten mL of a pTF-mCherry expressing Δ*pTF P. tricornutum* cell line grown in iron-deplete conditions (L1 medium with 7.5 nM total iron) were spun down, supernatant discarded, and pellet resuspended in 50 μL phosphate-buffered saline (PBS) (pH 7.4). 0.5 μL 10 mM MDY-64 stock (in DMSO; Thermo Fischer Scientific, Catalog #Y7536) was added, cells incubated for 10 min at RT, pelleted, and resuspended in 50 μL fresh PBS (pH 7.4). Cell suspension was prepared for imaging as follows: 5 μL was placed between a 1.5 mm microscope slide and a cover slip (this setup applies to all imaging experiments in the study). Imaging conditions: Leica TCS SP5 confocal microscope (Leica Microsystems, Wetzlar, Germany), argon laser strength set to 30%, 458 nm laser line at 50% maximum strength, emission window set to 477–517 nm (for visualizing MDY-64; MDY-64 excitation and emission maxima are 451 nm and 497 nm, respectively), 514 nm laser line at 50% maximum strength, emission window set to 620–640 nm (for visualizing mCherry; mCherry excitation and emission maxima are 587 nm and 610 nm, respectively).

## Protein expression analyses

### pTF-APEX2 detection

Cell pellets from 8 mL mid- to late-exponential phase WT or transconjugant *P. tricornutum* cultures were resuspended in 150 μL cell lysis buffer (50 mM Tris-HCl, 200 mM NaCl, 1 mM DTT, 1 mM PMSF, pH 8.5) and sonicated for 5 min (30 s on, 1 min off) with Bioruptor UCD-200TM (Diagenode, Liège, Belgium). The resulting cell lysates were centrifuged, total protein content in supernatants measured with Bradford Assay Kit (Thermo Fischer Scientific, Catalog #23236), and insoluble fractions resuspended in 150 μL cell lysis buffer. One μg of each soluble protein sample and 0.5 μL of each resuspended insoluble protein fraction were resolved on a NuPage 4–12% Bis-Tris 1.5 mm gel (Thermo Fischer Scientific, Catalog #NP0335BOX), wet transferred to polyvinylidene difluoride

(PVDF) membranes (Thermo Fischer Scientific, Catalog #LC2005), and visualized with WesternBreeze Chemiluminescent Kit, anti-rabbit (Thermo Fischer Scientific, Catalog #WB7106).

### pTF-mCherry detection

Cell pellets from 400 µL mid- to late-exponential phase WT or transconjugant *P. tricornutum* cultures were resuspended in 50 µL cell lysis buffer (50 mM Tris-HCl, 200 mM NaCl, 1 mM DTT, 1 mM PMSF, pH 8.5) and sonicated 15 min (30 s on, 1 min off) with Bioruptor UCD-200TM. One µL of each whole cell lysate was resolved on a NuPage 4–12% Bis-Tris 1.5 mm gel, wet transferred to PVDF membranes, and visualized with WesternBreeze Chemiluminescent Kit, anti-rabbit.

### pTF antibody details

Amino acid residues 32–223 served as the immunogen (N-terminal pTF region just downstream of the signal peptide). The antibody was produced in a rabbit. 1:10,000 dilution of 1.14 mg/mL antibody stock was used in this study for background-free results.

### Protein ladder

MagicMark XP Western Protein Standard (Thermo Fischer Scientific, Catalog #LC5602).

### Related resource

Turnšek J. 2017. HA tag enables highly efficient detection of heterologous proteins in *Phaeodactylum tricornutum* (*Pt*) exconjugants. *protocols.io*. DOI: dx.doi.org/10.17504/protocols.io.j7ncrme.

## Amplex UltraRed assay and resorufin imaging

### Amplex UltraRed assay

Five mL of triplicate WT or pTF-APEX2 expressing *P. tricornutum* cultures in mid- to late-exponential phase were incubated on ice for 5 min, spun down, supernatant discarded, pellet resuspended in 500 µL ice-cold PBS (pH 7.4), and transferred to microcentrifuge tubes. Cells were spun down again, resuspended in 200 µL ice-cold reaction buffer (50 µM Amplex UltraRed [AUR; Thermo Fischer Scientific, Catalog #A36006], 2 mM $H_2O_2$, in PBS [pH 7.4]), and incubated on ice for 15 min unless otherwise noted. Fifty µL supernatant was mixed with 50 µL PBS (pH 7.4) and resorufin fluorescence measured in a black microtiter plate with black bottom using Flexstation 3 microtiter plate reader (Molecular Devices, San Jose, CA; excitation: 544 nm, emission: 590 nm; resorufin excitation and emission maxima are 568 nm and 581 nm, respectively). Horseradish peroxidase (HRP) was always included as a positive assay control. Fluorescence was normalized to $OD_{750}$ of experimental *P. tricornutum* cultures. Amplex UltraRed was prepared as a 10 mM stock in DMSO and stored in 20 µL aliquots at −20℃. Three percent (w/w) $H_2O_2$ stock (Sigma-Aldrich, Saint Louis, MO, Catalog #323381–25 ML) was stored in 100 µL aliquots at −20℃.

### Resorufin imaging

WT and pTF-APEX2 expressing *P. tricornutum* cells after performing the Amplex UltraRed assay were imaged with Leica TCS SP5 confocal microscope using the following parameters: argon laser strength at 30%, 514 nm laser line at 50% maximum strength, resorufin emission window: 575–605 nm, autofluorescence emission window: 700–750 nm.

## Transmission electron microscopy (TEM)

### Part 1: Labeling

Five mL of triplicate WT or pTF-APEX2 expressing *P. tricornutum* cultures in mid- to late-exponential phase were spun down (4000 rpm, 4℃, 10 min) and fixed in 5 mL ice-cold 2% (w/v) paraformaldehyde (PFA) and 2% (v/v) glutaraldehyde in 0.15 M sodium cacodylate buffer (pH 7.4) for 30 min on ice. Cells were rinsed in 5 mL 0.15 M sodium cacodylate buffer (pH 7.4) five times for 3 min on ice, then once again in 5 mL 0.15 M sodium cacodylate buffer (pH 7.4) with 10 mM glycine for 3 min on ice. Cells were then treated with 25 mM 3,3'-diaminobenzidine (DAB; Sigma-Aldrich, Catalog #D8001) as follows: 5.36 mg DAB was dissolved in 1 mL 0.1 N HCl and sonicated for 45 min. Five mL of 0.3 M sodium cacodylate buffer (pH 7.4) was added to dissolved DAB, final volume adjusted to 10 mL with ddH$_2$O, solution filtered through a 0.22 µm syringe filter, and 3 µL 30% (w/w) $H_2O_2$

added for 3 mM final concentration. Cells were incubated in 5 mL of this reaction buffer for 15 min on ice, rinsed five times for 3 min in 5 mL 0.15 M sodium cacodylate buffer (pH 7.4), post-fixed in 2 mL 1% osmium tetroxide ($OsO_4$) (Electron Microscopy Sciences, Hatfield, PA, Catalog #19150) in 0.15 M sodium cacodylate buffer (pH 7.4) for 30 min on ice, and rinsed in 5 mL ice-cold $ddH_2O$ five times for 3 min. Cells were then resuspended in 300 µL melted 3% agar, poured onto a glass slide sitting on ice, left to solidify, and cut into small ~3×3×1 mm pieces which were transferred into scintillation vials with 10 mL ice-cold $ddH_2O$. All shorter spin down steps were done at 4000 rpm and 4°C for 1.5 min. All buffers were used ice-cold.

## Part 2: TEM sample preparation

Agar blocks with embedded *P. tricornutum* cells were fixed in 10 mL 2% (v/v) glutaraldehyde in $ddH_2O$ to crosslink agar, rinsed five times for 2 min in 5 mL ice-cold $ddH_2O$, and left incubating in 5 mL 2% uranyl acetate (Electron Microscopy Sciences, Catalog #22400) overnight at 4°C. Next morning, agar blocks were first dehydrated in the following ethanol series: 20%, 50%, 70%, 90%, 100% (on ice, 10 mL), 100%, 100% (at RT, 10 mL), then infiltrated with 10 mL 50% epoxy resin for ~1 hr. To prepare 20 mL 100% resin, Durcupan ACM mixture (Electron Microscopy Sciences, Catalog #14040) components were combined as follows: 11.4 g A (epoxy resin), 10 g B (964 hardener), 0.3 g C (964 accelerator), and 0.1 g D (dibutyl phthalate) in this exact order (for 50% resin 1 part 100% resin was combined with 1 part 100% ethanol). Agar blocks were transferred into 10 mL 100% resin for ~4 hr, then into fresh 10 mL 100% resin overnight, again into fresh 10 mL 100% resin for 4 hr the following morning before finally being poured into aluminum boats and left to polymerize in 60°C oven for at least 48 hr (over the weekend). Polymerized resins were detached from aluminum boats, agar blocks dense with cells cut out and glued to 'dummy' blocks by incubation in a 60°C oven for at least 15 min. Approximately 500-nm-thick sections were cut with an ultramicrotome (Leica Ultracut), stained with 1% toluidine blue, and observed under light microscope to make sure embedded cells were exposed. Approximately 100-nm-thick sections were then cut, placed on TEM grids, labeled, and saved until imaging. Imaging was performed with JEOL JEM-1200 (Japan Electron Optics Laboratory, Akishima, Tokyo, Japan) transmission electron microscope at 80 keV. Backscatter scanning electron microscopy was performed with Zeiss Merlin (Oberkochen, Germany) scanning electron microscope (SEM) at two keV by placing ~80-nm-thick sections on a silicon wafer and imaged with inverted contrast which gives a TEM-like image.

## Proximity-dependent proteomic mapping

### Part 1: Labeling

Twenty-five mL of quintuplicate WT and pTF-APEX2 expressing *P. tricornutum* cultures in mid- to late-exponential phase—cell density of all cultures just prior to harvest was ~$2×10^7$ cells/mL which corresponds to $OD_{750}$ ~0.4—were cooled on ice for 10 min and pelleted (4000 rpm, 4°C, 10 min). Supernatants were discarded, pellets resuspended in 0.5 mL ice-cold PBS (pH 7.4), transferred to microcentrifuge tubes, and spun down (4000 rpm, 4°C, 10 min). Cells were then resuspended in 0.5 mL ice-cold 1.2 M D-sorbitol in PBS (pH 7.4), supplemented with 2.5 mM biotin-phenol (Berry and Associates, Dexter, MI, Catalog #BT 1015), incubated on a tube rotator at 4°C for 90 min, supplemented with 1 mM $H_2O_2$, and incubated on a tube rotator at 4°C for another 20 min. Labeling reaction was quenched by washing cells twice (4000 rpm, 4°C, 5 min) with 0.5 mL ice-cold quenching solution (10 mM sodium ascorbate [VWR International, Radnor, PA, Catalog #95035–692], 5 mM Trolox [Sigma-Aldrich, Catalog #238813–5G], 10 mM sodium azide [VWR International, Catalog #AA14314-22] in PBS [pH 7.4]). Of quenched cell suspension, 50 µL was saved for a streptavidin blot. The remaining 450 µL was spun down (4000 rpm, 4°C, 10 min) and lysed in 250 µL cell lysis buffer (50 mM Tris-HCl, 200 mM NaCl, 1 mM DTT, 1 mM PMSF, pH 8.5) by sonication for 15 min (30 s on, 1 min off). Cell lysates were spun down at 4000 rpm and 4°C for 45 min and protein concentration in supernatants measured using Bradford Assay Kit. For streptavidin blot, 25 µL of saved quenched cells were first sonicated for 15 min (30 s on, 1 min off). A total of 2.5 µL of whole cell lysates and 1 ng biotinylated BSA positive control (Thermo Fischer Scientific, Catalog #29130) were then resolved on a NuPage 4–12% Bis-Tris 1.5 mm gel and wet transferred to a PVDF membrane. Membrane was washed twice with 15 mL PBST (PBS [pH 7.4] with 0.1% [v/v] Tween-20) for 5 min, left blocking overnight at RT and gentle shaking in PBST supplemented with 5% BSA (Sigma-Aldrich,

Catalog #A9647-100G), and washed once the next morning with 15 mL PBST for 5 min. It was then incubated for 1 hr at RT and gentle shaking in 15 mL PBST supplemented with 5% BSA and 1:15,000 HRP-conjugated streptavidin (Thermo Fischer Scientific, Catalog #S911). Finally, membrane was washed three times with 15 mL PBST supplemented with 5% BSA for 5 min and once with 15 mL PBST for 10 min after which it was visualized with SuperSignal West Dura Extended Duration reagent (Thermo Fischer Scientific, Catalog #34075) using C-DiGit Blot Scanner (Li-COR, Lincoln, NE).

### Protein ladder
WesternSure Pre-stained Chemiluminescent Protein Ladder (HRP-conjugated protein ladder) (Li-COR, Catalog #926–98000).

## Part 2: Pull-down and mass spectrometry
### Pull-down
Fifty µL streptavidin beads (Thermo Fischer Scientific, Catalog #88816) transferred to microcentrifuge tubes were pelleted in a magnetic rack (tubes were left standing for ~5 min to pellet fully). Supernatants were discarded and beads were washed twice with 1 mL ice-cold cell lysis buffer (50 mM Tris-HCl, 200 mM NaCl, 1 mM DTT, 1 mM PMSF, pH 8.5). A total of 360 µg proteins in 500 µL total volume (x µL supernatant from cell lysis with 360 µg proteins and 500-x µL cell lysis buffer) were incubated on a tube rotator overnight at 4°C. Streptavidin beads were washed to remove non-specific binders the next morning as follows: $2 \times 1$ mL cell lysis buffer, $1 \times 1$ mL 1 M KCl, $1 \times 1$ mL 0.1 M $Na_2CO_3$ (pH 11.5), $1 \times 1$ mL 2 M urea (pH 8.0) in 10 mM Tris-HCl, $2 \times 1$ mL cell lysis buffer, $2 \times 1$ mL PBS (pH 7.4). PBS after the final washing step was removed before storing beads at −80°C. Notes: All streptavidin beads collection steps were 5 min long. All the washing solutions were ice-cold. Microcentrifuge tubes were either very briefly vortexed (~2 s) or tapped by hand between each washing step to promote bead resuspension.

### On-bead digestion and TMT labeling
Samples were prepared as previously described (*Kalocsay, 2019*). Liquid reagents used were HPLC quality grade. Washed beads were resuspended in 50 µL of 200 mM EPPS (4-(2-hydroxyethyl)−1-piperazinepropanesulfonic acid) buffer (pH 8.5) and 2% (v/v) acetonitrile with 1 µL of 2 mg/mL lysil endoproteinase Lys-C stock solution (FUJIFILM Wako Pure Chemical Corporation, Richmond, VA, Catalog #125–05061), briefly vortexed, and incubated at 37°C for 3 hr. Fifty µL of trypsin stock (Promega, Madison, WI, Catalog #V5111) diluted 1:100 (v/v) in 200 mM EPPS (pH 8.5) was then added. After mixing, digests were incubated at 37°C overnight and beads were magnetically removed. Peptides were then directly labeled as follows: acetonitrile was added to 30% (v/v) concentration and peptides were labeled with TMT 10-plex reagent (Thermo Fisher Scientific, Catalog #90406) for 1 hr. Labeling reactions were quenched with hydroxylamine at a final concentration of 0.3% (v/v) for 15 min and 1% of labeled peptides was analyzed for label incorporation efficiency by mass spectrometry. After quenching, peptide solutions were first acidified with formic acid, trifluoroacetic acid (TFA) was then added to a concentration of 0.1% (v/v), and peptides were desalted by acidic $C_{18}$ solid phase extraction (StageTip). Labeled peptides were finally resuspended in 1% (v/v) formic acid and 3% (v/v) acetonitrile.

### Mass spectrometry
Data were collected with a MultiNotch MS3 TMT method (*McAlister et al., 2014*) using an Orbitrap Lumos mass spectrometer coupled to a Proxeon EASY-nLC 1200 Liquid Chromatography (LC) system (both Thermo Fisher Scientific). The used capillary column was packed with $C_{18}$ resin (35 cm length, 100 µm inner diameter, 2.6 µm Accucore matrix [Thermo Fisher Scientific]). Peptides were separated for 3 or 4 hr over acidic acetonitrile gradients by LC prior to mass spectrometry (MS) analysis. Data from two 4 hr runs and one 3 hr run were recorded and combined. After an initial $MS^1$ scan (Orbitrap analysis; resolution 120,000; mass range 400–1400 Th), $MS^2$ analysis used collision-induced dissociation (CID, CE = 35) with a maximum ion injection time of 150–300 ms and an isolation window of 0.5 m/z. In order to obtain quantitative information, $MS^3$ precursors were fragmented by high-energy collision-induced dissociation (HCD) and analyzed in the Orbitrap at a resolution of 50,000 at 200 Th. Further details on LC and MS parameters and settings were described recently (*Paulo et al., 2016*). The mass spectrometry proteomics data have been

deposited to the ProteomeXchange Consortium via the PRIDE partner repository (*Perez-Riverol et al., 2019*) with the dataset identifier PXD018022.

## pTF-mCherry and MS hit-EYFP co-expression and imaging

Sequence-validated episomes with pTF-mCherry and MS hit-EYFP expression cassettes were conjugated into WT *P. tricornutum* cells, resulting transconjugants genotyped, and screened for fluorescence. Five µL of fluorescent transconjugant cell lines in mid- to late-exponential phase were imaged with settings that minimized cross-channel bleed-through: argon laser strength at 30%, 514 nm laser line at 50% maximum strength, mCherry emission window: 620–640 nm, EYFP emission window: 520–540 nm, autofluorescence emission window: 700–750 nm.

## pTF.CREG1 protein expression conditions

Small-scale expression testing for the Δ31_pTF.CREG1-His6 (and all other pTF.CREG1 truncations not described here) was performed as follows: an overnight culture of BL21 *E. coli* cells carrying the PtpBAD-Δ31_pTF.CREG1-CTHF (=pJT_Δ31_pTF.CREG1-His6) expression vector was used to inoculate 50 mL of Terrific Broth (TB) supplemented with tetracycline (10 µg/mL). Cultures were grown in a shaking incubator (37°C, 200 rpm) until $OD_{600}$ of 0.4–0.6 was reached. Incubator temperature was then lowered to 18°C and flasks were allowed to adjust to this temperature for about 30 min before arabinose was added to a final concentration of 0.5% (w/v). Growth at 18°C was continued overnight (12–18 hr) after which 10 mL of cultures were harvested by centrifugation at 6000 g for 10 min. Large-scale Δ31_pTF.CREG1-His6 expression was performed as above, but at 2 L total volume. Following arabinose induction and overnight growth of the 2 L culture, 500 mL was harvested for further processing.

## pTF.CREG1 purification conditions

All purifications were performed using cobalt, $Co^{2+}$, TALON Metal Affinity Resin (Takara Bio, Catalog #635502) which binds His6-tagged proteins with high affinity. Pellets from small scale expression testing (10 mL culture) were resuspended in 800 µL of lysis buffer (50 mM $NaH_2PO_4$ [pH 7.5], 500 mM NaCl, 0.1% [v/v] Triton X-100, 10 mM imidazole, 1 mg/mL lysozyme, and 10 µM β-mercaptoethanol) and subjected to microtip sonication on ice until lysis was complete. Lysates were then clarified by centrifugation (10 min, 15,000 g, 4°C) and supernatants (700 µL) were set aside. For each purification, 25 µL of TALON resin was equilibrated by washing three times with 10 volumes of lysis buffer and pelleting by centrifugation after each wash (30 s, 3000 g, RT). After third wash, 25 µL of lysis buffer was added to the resin to make a 50 µL slurry. Supernatants from previously clarified cell lysates were then added to the TALON resin slurry and incubated for 1 hr at RT with end-over-end mixing. Following the 1 hr incubation, the resin was pelleted by centrifugation (30 s, 3000 g, RT) and washed three times with 10 volumes of wash buffer (50 mM $NaH_2PO_4$ [pH 7.5], 500 mM NaCl, 30 mM imidazole). Proteins were eluted with 50 uL of elution buffer (50 mM $NaH_2PO_4$ [pH 7.5], 500 mM NaCl, 250 mM imidazole, 10% glycerol) and were either subjected to immediate SDS-PAGE electrophoresis or stored at −80°C.

Pellets from large-scale expression (500 mL harvested bacterial culture) were processed with a few modifications. For lysis, cell pellets were resuspended in 20 mL of lysis buffer. For purification, 1 mL of TALON resin (i.e. 2 mL of equilibrated slurry) was used in combination with approximately 19 mL of clarified cell lysate which was allowed to incubate overnight at 8°C with end-over-end mixing. Final protein elution was performed using 5 mL of elution buffer. Purified protein was subjected to SDS-PAGE electrophoresis and concentrated using an Amicon Ultra-15 10 kDa cutoff concentrator (Millipore Sigma, Burlington, MA, Catalog #UFC901024). Following the concentration step, buffer exchange into 20 mM HEPES (pH 8.0), 10% glycerol, 300 mM KCl was performed using a PD-10 desalting column (GE Healthcare, Chicago, IL, Catalog #17-0851-01). Protein was then concentrated again and total protein content was determined by the Bradford method using the Protein Assay Dye Reagent (Bio-Rad, Hercules, CA, Catalog #5000006).

## Flavin reduction assays

To test for flavin reductase activity, we set up the following enzyme assay conditions in 100 uL final volume: 50 mM Tris-HCl (pH 7.5), 100 mM KCl, 10% glycerol, 30 µM flavin mononucleotide (FMN),

30 μM riboflavin, and 250 μM NADPH (*Coves and Fontecave, 1993*). Assays were started by the addition of 50 μg of pTF.CREG1, BSA, or Milli-Q water for no enzyme controls. Oxidation of NADPH was measured by the decrease in absorbance at 320 nm on a Flexstation 3 microtiter plate reader and the signal for each reaction was tracked over a period of 60 min. All assays were conducted in triplicate.

## Bioinformatic and data analyses

### Mass spectrometry data analysis

Peptide-spectrum matches used a SEQUEST (v. 28, rev. 12) algorithm (*Eng et al., 1994*). Data were searched against a size-sorted forward and reverse database of the *Phaeodactylum tricornutum* proteome (strain CCAP 1055/1, UniProt reference proteome UP000000759) with added common contaminant proteins and the pTF-APEX2 fusion protein sequence. Spectra were first converted to mzXML and searches were then performed using a mass tolerance of 50 ppm for precursors and a fragment ion tolerance of 0.9 Da. For the searches, maximally two missed cleavages per peptide were allowed. We searched dynamically for oxidized methionine residues (+15.9949 Da) and applied a target decoy database strategy. A false discovery rate (FDR) of 1% was set for peptide-spectrum matches following filtering by linear discriminant analysis (LDA) (*Beausoleil et al., 2006*; *Huttlin et al., 2010*). The FDR for final collapsed proteins was 1%. Quantitative peptide information was derived from $MS^3$ scans. Quant tables were generated with the following filter criteria: $MS^2$ isolation specificity of >70% for each peptide and a sum of TMT signal-to-noise (s/n) of >200 over all channels per peptide. Quant tables were exported to Excel and further processed therein. Details of the TMT intensity quantification method and additional applied search parameters were described previously (*Paulo et al., 2016*). Scaled proteomic data were subjected to two-way hierarchical clustering (Ward's method) using JMP software package (SAS Institute). Volcano plot with $log_2$-transformed average APEX2/WT ratios and associated p values was made in R using ggplot2 data visualization package (*R Development Core Team, 2013*). Gene IDs corresponding to protein hits were inferred using Ensembl Protists Phatr3 *P. tricornutum* genomic database.

### Protein feature identification

Protein lengths, molecular weights, and isoelectric points (pI) were determined with ProtParam (*Gasteiger et al., 2005*). Signal peptides and transmembrane regions were identified with SignalP 4.1 (*Petersen et al., 2011*) and TMHMM Server v. 2.0 (*Krogh et al., 2001*), respectively. Protein localizations were predicted with a combination of tools: TargetP 1.1 (*Emanuelsson et al., 2000*), SignalP 4.1, and ASAFind (*Gruber et al., 2015*) version 1.1.7. All putative chloroplastic localizations in the study mean that a protein was predicted to be chloroplastic by ASAFind (state-of-the-art plastidial protein localization prediction tool for diatoms); chloroplast localization prediction confidences are noted. Low confidence prediction by ASAFind means that a protein satisfies the following filtering criteria: (1) it contains a signal peptide as detected by SignalP 4.1, (2) +1 position of ASAFind predicted cleavage site is phenylalanine (F), tryptophan (W), tyrosine (Y), or leucine (L) (making the protein 'potentially plastid targeted'; it was tyrosine in pTF.CREG1), and (3) one or both of the following is false: the ASAFind predicted cleavage site coincides with the SignalP 4.1 prediction and the transit peptide score is higher than 2 (the latter is false for pTF.CREG1: transit peptide score was ~0.81). Peroxidase class prediction was done in RedoxiBase (*Savelli et al., 2019*). N-glycosylation and GalNAc-type O-glycosylation predictions were performed with NetNGlyc 1.0 (*Gupta et al., 2004*) and NetOGlyc 4.0 (*Steentoft et al., 2013*), respectively.

### Amino acid sequence alignments

APEX2-like *P. tricornutum* peroxidases were aligned with Clustal Omega (*Sievers et al., 2011*) using the default settings and CLC Sequence Viewer 7.7 (QUIAGEN) using the following parameters: Gap open cost: 10.0, Gap extension cost: 1.0, End gap cost: As any other, Alignment: Very accurate (slow). Putative substrate-binding loops in these peroxidases were evaluated using the Clustal Omega alignment. Conserved motifs in amino acid sequence alignments underlying the phylogenetic trees were displayed in CLC Sequence Viewer 7.7.

## Phylogenetic analyses

Homologs of respective *P. tricornutum* proteins were retrieved from the non-redundant National Center for Biotechnology Information (NCBI) and the Marine Microbial Eukaryote Transcriptome Sequencing Project (MMETSP) databases using the BLASTP algorithm (e-value threshold set to $e^{-15}$) (*Altschul et al., 1990*). The BLAST search retrieved only a handful of homologs for pTF.ap1 and no homologs for pTF.CatCh1 even among closely related diatoms. Therefore, more sensitive HMMER (*Eddy, 1998*) searches were employed to extend the datasets, which were afterward aligned using the Localpair algorithm as implemented in MAFFT (*Katoh et al., 2002*). Ambiguously aligned regions, regions composed mostly of gaps, and short fragments were manually removed in SeaView 4 (*Gouy et al., 2010*). For each alignment, the maximum likelihood analysis was carried out in IQ-TREE (*Nguyen et al., 2015*) under the best-fitting substitution matrix as inferred by the built-in model finder. Branching support was estimated using 'thorough' non-parametric bootstrap analysis from 500 replicates in IQ-TREE.

## Identification of disordered protein regions

pTF.CatCh1 amino acid sequence was analyzed for the presence of disordered protein regions with PONDR (Predictor of Naturally Disordered Regions) VSL2 predictor (*Peng et al., 2006*) and IUPred2a long disorder prediction type (*Mészáros et al., 2018*).

## Homology modeling of pTF.CREG1

Full-length pTF.CREG1 amino acid sequence was used as an input for Phyre2, an online protein structure prediction server (*Kelley et al., 2015*). Normal modeling mode was selected. Human CREG1 (PDB ID: 1XHN) was identified as the best modeling template (human CREG1 is ~37% identical to pTF.CREG1 as inferred from their Clustal Omega alignment). pTF.CREG1 was modeled with >90% confidence across 182 modeled amino acid residues (Pro43–Lys224) indicating the predicted structure is representative of the actual structure. pTF.CREG1 homology model was aligned with human CREG1 monomer in UCSF Chimera version 1.11.1 using MatchMaker tool with default settings (*Pettersen et al., 2004*).

## Other data analyses

Amplex UltraRed assay and transcriptomic data were plotted in R using ggplot2 data visualization package (*R Development Core Team, 2013*). Flavin reduction assay data were plotted in Excel.

## Acknowledgements

We thank Andrea Thor and Mason Mackey for help with electron microscopy sample preparation and imaging; Marian Kalocsay for assistance with mass spectrometry experiments and data analyses; Pardis Gholami and Hong Zheng for laboratory assistance; Jeffrey D Martell, Jeffrey B McQuaid, Tyler H Coale, and Sarah R Smith for fruitful discussions. This project was supported by the Gordon and Betty Moore Foundation grants GBMF4958 (JT), GBMF3828 (AEA), and GBMF5006 (AEA), the National Science Foundation grants NSF-OCE-1756884 (AEA) and NSF-MCB-1818390 (AEA), the United States Department of Energy Genomics Science program grant DE-SC0018344 (AEA), the Czech Science Foundation grant 21-03224S (MO), and ERDF/ESF Centre for Research of Pathogenicity and Virulence of Parasites grant No.CZ.02.1.01/0.0/0.0/16_019/0000759 (MO). JKB was supported by the National Institutes of Health (NIH) Ruth L Kirschstein Predoctoral Individual National Research Service Award F31 1F31ES030613-01. National Center for Microscopy and Imaging Research (NCMIR) is supported by the National Institute for General Medical Sciences (NIGMS) of the National Institutes of Health (NIH).

## Additional information

### Funding

| Funder | Grant reference number | Author |
| --- | --- | --- |
| Gordon and Betty Moore | GBMF3828 | Andrew Ellis Allen |

| Foundation | | |
|---|---|---|
| Gordon and Betty Moore Foundation | GBMF5006 | Andrew Ellis Allen |
| National Science Foundation | NSF-OCE-1756884 | Andrew Ellis Allen |
| National Science Foundation | NSF-MCB-1818390 | Andrew Ellis Allen |
| Biological and Environmental Research | DE-SC0018344 | Andrew Ellis Allen |
| Gordon and Betty Moore Foundation | GBMF4958 | Jernej Turnšek |
| National Institutes of Health | 1F31ES030613-01 | John K Brunson |
| Czech Science Foundation | 21-03224S | Miroslav Oborník |
| European Regional Development Fund | CZ.02.1.01/0.0/0.0/16_019/0000759 | Miroslav Oborník |
| National Institutes of Health | F31 1F31ES030613-01 | John K Brunson |

The funders had no role in study design, data collection and interpretation, or the decision to submit the work for publication.

## Author contributions

Jernej Turnšek, Conceptualization, including proposing the idea to implement APEX2 in diatoms, Data curation, Formal analysis, Funding acquisition, Investigation, Methodology, Project administration, Resources, Supervision, Validation, Visualization, Writing – original draft preparation, Writing – review and editing; John K Brunson, Vincent A Bielinski, Conceptualization, Data curation, Formal analysis, Funding acquisition, Investigation, Methodology, Project administration, Resources, Supervision, Validation, Visualization, Writing – review and editing; Maria del Pilar Martinez Viedma, Conceptualization, Resources, Data curation, Formal analysis, Supervision, Validation, Investigation, Visualization, Methodology, Writing - review and editing, performed expressing and purifying new, cleaner, pTF.CREG1 batches for biochemical assays; Thomas J Deerinck, Conceptualization, Resources, Data curation, Formal analysis, Supervision, Validation, Investigation, Visualization, Methodology; Aleš Horák, Miroslav Oborník, Conceptualization, Resources, Data curation, Formal analysis, Supervision, Validation, Investigation, Visualization; Andrew Ellis Allen, Conceptualization, Resources, Data curation, Formal analysis, Supervision, Funding acquisition, Validation, Investigation, Visualization, Methodology, Project administration, Writing - review and editing

## Author ORCIDs

Jernej Turnšek https://orcid.org/0000-0002-9056-3565
Maria del Pilar Martinez Viedma https://orcid.org/0000-0002-8885-5008
Andrew Ellis Allen https://orcid.org/0000-0001-5911-6081

## Decision letter and Author response

Decision letter https://doi.org/10.7554/eLife.52770.sa1
Author response https://doi.org/10.7554/eLife.52770.sa2

# Additional files

## Supplementary files

• Supplementary file 1. Supplementary tables. Table S1. Predicted endogenous biotinylated proteins are present at similar levels in WT and pTF-APEX2 proteomic samples. Table S2: Features of the three proteins co-expressed from a gene cluster on chromosome 20. Table S3: Molecular cloning primers. Table S4: Constructed *Phaeodactylum tricornutum* episomes and *Escherichia coli* vectors with expression cassette details and fusion protein amino acid sequences.

• Transparent reporting form

## Data availability

All data generated or analyzed during this study are included in the manuscript and supporting files.

The following dataset was generated:

| Author(s) | Year | Dataset title | Dataset URL | Database and Identifier |
|---|---|---|---|---|
| Turnšek J, Allen AE | 2021 | Quantitative proximity proteomics suggests a phytotransferrin-mediated cell surface-to-chloroplast iron trafficking axis in marine diatoms | https://www.ebi.ac.uk/pride/archive/projects/PXD018022 | PRIDE, PXD018022 |

The following previously published datasets were used:

| Author(s) | Year | Dataset title | Dataset URL | Database and Identifier |
|---|---|---|---|---|
| Bowler C, Allen AE, Badger JH, Grimwood J, Jabbari K, Kuo A, Maheswari U, Martens C, Maumus F, Otillar RP, Rayko E, Salamov A, Vandepoele K, Beszteri B, Gruber A, Heijde M, Katinka M, Mock T, Valentin K, Verret F, Berges JA, Brownlee C, Cadoret JP, Chiovitti A, Choi CJ, Coesel S, De Martino A, Detter JC, Durkin C, Falciatore A, Fournet J, Haruta M, Huysman MJ, Jenkins BD, Jiroutova K, Jorgensen RE, Joubert Y, Kaplan A, Kroger N, Kroth PG, La Roche J, Lindquist E, Lommer M, Martin-Jezequel V, Lopez PJ, Lucas S, Mangogna M, McGinnis K, Medlin LK, Montsant A, Oudot-Le Secq MP, Napoli C, Obornik M, Parker MS, Petit JL, Porcel BM, Poulsen N, Robison M, Rychlewski L, Rynearson TA, Schmutz J, Shapiro H, Siaut M, Stanley M, Sussman MR, Taylor AR, Vardi A, von Dassow P, Vyverman W, Willis A, Wyrwicz LS, Rokhsar DS, Weissenbach J, Armbrust EV, Green BR, | 2008 | Phaeodactylum tricornutum proteome | https://www.uniprot.org/proteomes/UP000000759 | UniProt, UP000000759 |

| | | | | |
|---|---|---|---|---|
| Van de Peer Y, Grigoriev IV | | | | |
| Rastogi A, Maheswari U, Dorrell RG, Vieira FRJ, Maumus F, Kustka A, McCarthy J, Allen AE, Kersey P, Bowler C, Tirichine L | 2018 | Phaeodactylum tricornutum genome annotation 3 (Phatr3) | http://protists.ensembl.org/Phaeodactylum_tricornutum/Info/Index?db=core | Ensembl, ASM15095v2 |
| Keeling PJ, Burki F, Wilcox HM, Allam B, Allen EE, Amaral-Zettler LA, Armbrust EV, Archibald JM, Bharti AK, Bell CJ, Beszteri B, Bidle KD, Cameron CT, Campbell L, Caron DA, Cattolico RA, Collier JL, Coyne K, Davy SK, Deschamps P, Dyhrman ST, Edvardsen B, Gates RD, Gobler CJ, Greenwood SJ, Guida SM, Jacobi JL, Jakobsen KS, James ER, Jenkins B, John U, Johnson MD, Juhl AR, Kamp A, Katz LA, Kiene R, Kudryavtsev A, Leander BS, Lin S, Lovejoy C, Lynn D, Marchetti A, McManus G, Nedelcu AM, Menden-Deuer S, Miceli C, Mock T, Montresor M, Moran MA, Murray S, Nadathur G, Nagai S, Ngam PB, Palenik B, Pawlowski J, Petroni G, Piganeau G, Posewitz MC, Rengefors K, Romano G, Rumpho ME, Rynearson T, Schilling KB, Schroeder DC, Simpson AG, Slamovits CH, Smith DR, Smith GJ, Smith SR, Sosik HM, Stief P, Theriot E, Twary SN, Umale PE, Vaulot D, Wawrik B, Wheeler GL, Wilson WH, Xu Y, Zingone A, Worden AZ | 2014 | Marine Microbial Eukaryote Transcriptome Sequencing Project (MMETSP) database | https://www.ncbi.nlm.nih.gov/bioproject/248394 | NCBI BioProject, PRJNA231566 |

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

# Appendix 1

## Glossary

**AA/AAs**: amino acid residue/amino acid residues
**APEX**: engineered ascorbate peroxidase (APX)
**CREG**: cellular repressor of E1A-stimulated genes
**DAB**: 3,3'-diaminobenzidine
**EYFP**: enhanced yellow fluorescent protein
**Fe'**: dissolved labile iron (all unchelated iron species)
**FMN**: flavin mononucleotide
**HNLC**: high-nutrient, low-chlorophyll
**ISIP**: iron starvation induced protein
**MS**: mass spectrometry
**PBS**: phosphate-buffered saline
**pTF**: phytotransferrin
**RT**: room temperature
**TEM**: transmission electron microscopy
**TM**: transmembrane
**TMT**: tandem mass tag
**UTR**: untranslated region
**V-ATPase**: vacuolar-type $H^+$-ATPase
**WT**: wild type

