## [Decision Letter]

**Acceptance summary:**

Your study impacts the fields of microbial iron homeostasis and broadens the understanding of cofactor trafficking in chloroplast-containing eukaryotes. Although many of the proteins identified in this study have yet to be experimentally characterized, the results presented provide an important starting point for further in detail characterization of iron assimilation and trafficking.

**Decision letter after peer review:**

Thank you for submitting your article "Phytotransferrin endocytosis mediates a direct cell surface-to-chloroplast iron trafficking axis in marine diatoms" for consideration by *eLife*. Your article has been reviewed by two peer reviewers, and the evaluation has been overseen by Christian Hardtke as the Senior and Reviewing Editor.

The reviewers have discussed the reviews with one another and the Reviewing Editor has drafted this decision to help you prepare a revised submission.

There are a few essential revisions that are required to make the paper acceptable for publication:

– Please improve the presentation of the microscopy data. This would require further, high quality images, to provide additional evidence for localisation. Ideally, this would include further analysis of distribution of markers in relation to chloroplasts, vesicles, cytosol etc. Z-stacks might help here to establish unambiguous localisation, and quantitative data might help to bolster the claims.

– As it is written now, the paper is at times very speculative and gives the impression that the authors try to fit the data to their preferred model. Please reduce the level of speculation throughout the manuscript and remove discussion that is not directly relevant to the main findings.

– Related to the above, please modify the title, which, in the editor’s opinion, oversells the results of the paper as it stands now.

Finally, please address the comments in the individual reviews below point-by-point in your revised manuscript.

Reviewer #1:

The manuscript by Turnek et al. addresses an important aspect of Fe acquisition and builds on the previous work of the lead authors on this topic. They applied APEX-2 genetic engineering "proximity proteomics" that has previously been developed in animal and yeast to localize proteins that from close associations with diatom pTF at both light microscopy and EM levels of resolution. The work represents a comprehensive method development and demonstrates the strength of this and related approaches. It represents an important contribution to the field of phytoplankton research and the described method should have wide applicability in dissecting components of complex pathways.

The authors identified a number of proteins encoded by a gene cluster on chromosome 20 and carried out partial in silico, localization and biochemical studies on 3 of them. A key finding is the localization of pTF-protein associations in close proximity to the chloroplast, suggesting a possible transport route from the external medium to the photosynthetic apparatus. The manuscript is somewhat rambling and speculative in places. While the method development and quality of the data appears to be high, there is a lot of speculation based on in silico predictions, which I feel could be toned down. While a number of novel pTF associations are shown the manuscript rightly leaves open many questions concerning their precise function that can only be addressed in future by further biochemical, localization and genetic manipulation approaches.

1) Figure 5. Why is the distribution of pTF-mCherry (close to chloroplast) different to that shown in Figure 2 (discrete vesicles and cytoplasm)? According to the cartoon in Figure 9, pTF should be present in all of these compartments. This suggests some cell-cell variability in the distribution of pTF which should be resolved by showing a wider range of images and if possible applying statistical tests (e.g. quantify co-occurrence with "ferrosome" compartment close to the chloroplast).

2) Discussion: What is the significance of lack of inhibition of exogenous APEX2 peroxidase on ice?

3) Discussion: There is some uncertainty about the orientation of APEX2 with respect to the vesicle membrane. However, the data of Figure 2—figure supplement 1, showing cytoplasmic accumulation of resorufin would suggest a cytosolic orientation.

4) There is speculation about the role of FBAC5, which is stated to be known to be peroxisome localized but also with a function in regulation of vesicle H^+^-ATPase activity. Some more detailed localization studies, as shown with pTF.CREGr and pTFCatCh1 would have been informative here.

5) The speculation in the Discussion that carbonate may itself be a cargo, leading to increased CO2 levels around RuBisCO seems unlikely, considering the amounts of carbonate co-ordinated with Fe in relation to the total inorganic carbon fluxes that fuel C fixation, given that Fe is a trace element performing a catalytic function. The direct influence of the pTF pathway on total cell CO2 fluxes is likely to be very small.

6) Conclusions and Outlook: This section seems to be irrelevant to the focus of the manuscript. Indeed the whole Conclusions and Outlook, which reads like a thesis summary section, doesn't seem to fit or add much to the overall focus of the manuscript. Reference to the use of TurbolD (also in the Discussion) begs the question of why this approach was not used in the first place. Again, I would recommend that the manuscript could be considerably focussed and shortened, focussing on the main key findings and reducing the overall levels of speculation.

Reviewer #2:

I enjoyed reading the results of this study and am excited about this new chapter in understanding metal transport to complex plastids.

1) The most exciting aspect of this study, surface-to-chloroplast iron trafficking, is also the least supported aspect of this study. The results are exciting but preliminary with regards to such a significant discovery. For instance, most of the hypothesized members in the "trafficking axis" have yet to be functionally characterized in algae and iron binding by the proposed chaperone, which has the most convincing chloroplast proximity, was not demonstrated. I suggest modifying the title of the manuscript to something like "Phytotransferrin endocytosis and the identification of a putative surface-to-chloroplast iron trafficking pathway".

2) Unless I didn't find them in the supplementary material, only a single, 2D cell image is presented for each fusion construct. Multiple representative images should be provided. Also, because of the vacuole, the chloroplast, other organelles, and endosome-like vesicles could appear in very close proximity. A z-stack may help disambiguate the localization.

---

## [Author Response]

There are a few essential revisions that are required to make the paper acceptable for publication:– Please improve the presentation of the microscopy data. This would require further, high quality images, to provide additional evidence for localisation. Ideally, this would include further analysis of distribution of markers in relation to chloroplasts, vesicles, cytosol etc. Z-stacks might help here to establish unambiguous localisation, and quantitative data might help to bolster the claims.

We include additional three representative microscopy images per protein from the colocalization strains in Figure 5B (Figure 5—figure supplement 3). These micrographs further establish the described and discussed pTF.CREG1 and pTF.CatCh1 localization “phenotypes”. We agree that additional markers (e.g., Rab5a (Phatr3_J51511) and Rab11 (Phatr3_J31160) for early and late endosomes, respectively, and/or additional subcellular markers from Liu et al., 2016), Z-stacks, and quantitative data would help establish unambiguous localization, however these experiments were not feasible given the Spring stay-at-home order and the lead author’s move away from San Diego.

– As it is written now, the paper is at times very speculative and gives the impression that the authors try to fit the data to their preferred model. Please reduce the level of speculation throughout the manuscript and remove discussion that is not directly relevant to the main findings.

Additional enzymatic control experiments and structural comparison insights (Figure 6—figure supplement 2 and Figure 6—figure supplement 3) lead us to propose a non-enzymatic role for pTF.CREG1. This is in contrast to our initial submission where we hint at a possibility that pTF.CREG1 is an FMN-dependent reductase.

We significantly streamlined the manuscript’s narrative. Our model (Figure 8) now highlights only proteins that we either colocalized with pTF (pTF.CREG1 and pTF.CatCh1) or that have previously been shown to be critical in intracellular iron homeostasis (FBP1 and ISIP1) in *P. tricornutum*. The putative V-ATPase subunit is the sole exception. Finally, while we look to the metazoan transferrin cycle for inspiration, we now provide the readers with what we think is a more balanced discussion on these *P. tricornutum* proteins in relation to it.

– Related to the above, please modify the title, which, in the editors' opinion, oversells the results of the paper as it stands now.

We think our new title “Proximity proteomics in a marine diatom reveals a putative cell surface-to-chloroplast iron trafficking pathway” (111 characters with spaces) appropriately tones down the biological insights that could be gleaned from our work while remaining just slightly provocative. Given that this is the first time proximity proteomics was applied to a marine diatom, we consider highlighting this in the title is warranted. All in all, we believe keeping the title as is will broaden the target audience.

Alternative titles we are considering are as follows:

“Identification of a putative cell surface-to-chloroplast iron trafficking pathway in a marine diatom using proximity-dependent proteomic mapping” (144 characters with spaces; in case a 120-character limit for the title is not strictly enforced, we would like to consider this to be our title)

“Proximity-dependent proteomic mapping in a marine diatom reveals a putative cell surface-to-chloroplast iron trafficking pathway” (128 characters with spaces).

Finally, please address the comments in the individual reviews pasted point-by-point in your revised manuscript.

All individual reviews are addressed below.

Reviewer #1:[…] The authors identified a number of proteins encoded by a gene cluster on chromosome 20 and carried out partial in silico, localization and biochemical studies on 3 of them. A key finding is the localization of pTF-protein associations in close proximity to the chloroplast, suggesting a possible transport route from the external medium to the photosynthetic apparatus. The manuscript is somewhat rambling and speculative in places. While the method development and quality of the data appears to be high, there is a lot of speculation based on in silico predictions, which I feel could be toned down. While a number of novel pTF associations are shown the manuscript rightly leaves open many questions concerning their precise function that can only be addressed in future by further biochemical, localization and genetic manipulation approaches.

We agree that certain aspects of our initial manuscript were too speculative and are thus reduced or completely omitted in this revised version. We now focus our Results and Discussion on proteins that we either colocalized with pTF or that have previously been shown to be critical in intracellular iron homeostasis in *P. tricornutum*. This streamlining is also evident in our model (Figure 8). While we do discuss in silico findings, the corresponding figures are now largely in the supplement. We keep pTF.CatCh1 alignment with ISIP2b in the main text (Figure 7B) as we think highlighting the conserved cysteine-based motifs is important given their known role in metal-binding and redox active proteins. pTF.CREG1 enzymatic assay results and structural comparisons (Figure 6—figure supplement 2 and Figure 6—figure supplement 3) are in sync with what is known or proposed for human CREG1.

1) Figure 5. Why is the distribution of pTF-mCherry (close to chloroplast) different to that shown in Figure 2 (discrete vesicles and cytoplasm)? According to the cartoon in Figure 9, pTF should be present in all of these compartments. This suggests some cell-cell variability in the distribution of pTF which should be resolved by showing a wider range of images and if possible applying statistical tests (e.g. quantify co-occurrence with "ferrosome" compartment close to the chloroplast).

pTF does indeed exhibit some cell-to-cell variability. While pTF-YFP in Morrissey et al., 2015 appears to be localized exclusively to the cell surface (Figure 2D, indirectly also Figure 2E), work by McQuaid et al., 2018 (Figure 2D) and work presented in this study suggest pTF-mCherry is present in intracellular vesicles and in chloroplast vicinity (in addition to cell surface), and that this localization pattern is not contingent on deplete iron environment but is rather a constitutive cellular feature. We’d like to emphasize that Figure 2D in McQuaid et al., 2018 shows three distinct localization patterns for pTF: cell surface, what appear to be intracellular vesicles, and chloroplast-associated. All three have been observed in our work as well. Some cell-to-cell variability likely reflects the fact that none of the experimental cultures were synchronized and as such cells in all cell cycle stages were imaged. Localization data in Figure 5B is now supported by a new supplementary figure (Figure 5—figure supplement 3).

2) Discussion: What is the significance of lack of inhibition of exogenous APEX2 peroxidase on ice?

The following two sentences in the revised manuscript address this question:

“In our Amplex UltraRed assay experiments, an order of magnitude higher signal-to-noise ratio (i.e., resurofin signal in pTF-APEX2 expressing versus WT *P. tricornutum* cells) was detected when live cells were reacted on ice as opposed to on room temperature, suggesting endogenous *P. tricornutum* peroxidases, but not exogenous APEX2, are largely inhibited on ice. This is notable as it indicates that performing APEX2 assays on ice may represent a generalizable experimental strategy for mesophilic microeukaryotic phytoplankton model systems.”

3) Discussion: There is some uncertainty about the orientation of APEX2 with respect to the vesicle membrane. However, the data of Figure 2—figure supplement 1, showing cytoplasmic accumulation of resorufin would suggest a cytosolic orientation.

We would first like to bring your attention to a sentence that explicitly says what we predict in vivo APEX2 orientation is:

“Considering the predicted pTF domains (Figure 2—figure supplement 1A), APEX2 was likely facing the cytosol at the cell surface and once internalized into vesicles.”

We acknowledge the possibility that cytosolic resorufin accumulation indicates cytosolic APEX2 orientation:

“Resorufin was also directly visualized by confocal microscopy and a strong cytosolic signal not tightly localized to the expected site of origin, similar to previous reports (Martell et al., 2012), was observed (Figure 2—figure supplement 1B), perhaps indirectly supporting our predicted APEX2 orientation.”

4) There is speculation about the role of FBAC5, which is stated to be known to be peroxisome localized but also with a function in regulation of vesicle H^+^-ATPase activity. Some more detailed localization studies, as shown with pTF.CREGr and pTFCatCh1 would have been informative here.

We do not state FBAC5 is peroxisome-localized anywhere in our manuscript (i.e., neither in the first manuscript iteration nor in this revised one). Allen et al., 2012 show that FBAC5 is pyrenoid-localized. Given that FBAs can moonlight (i.e., perform functions other than their canonical one) (Boukouris et al., 2016) and considering some previous work suggesting FBAs’ role in membrane trafficking-related processes (Merkulova et al., 2011), we believe it is plausible that FBAC5 has an additional function away from chloroplast interior, perhaps in pTF internalization and subsequent endocytic trafficking. While we keep a paragraph on FBAC5 in the Discussion, it is no longer in our model.

We fully agree that additional localization and co-localization studies would be informative (not only for FBAC5, but for many other interesting MS hits in our study), however these additional experiments are beyond the scope of our current work.

5) The speculation in the Discussion that carbonate may itself be a cargo, leading to increased CO2 levels around RuBisCO seems unlikely, considering the amounts of carbonate co-ordinated with Fe in relation to the total inorganic carbon fluxes that fuel C fixation, given that Fe is a trace element performing a catalytic function. The direct influence of the pTF pathway on total cell CO2 fluxes is likely to be very small.

We agree and this section has been omitted.

6) Conclusions and Outlook: This section seems to be irrelevant to the focus of the manuscript. Indeed the whole Conclusions and Outlook, which reads like a thesis summary section, doesn't seem to fit or add much to the overall focus of the manuscript.

We agree and the “Conclusions and Outlook” section has been omitted. Instead, we end the manuscript with three paragraphs of decreasing specificity/increasing breadth: we introduce the model (Figure 8) given the presented and discussed data, and outline future steps that could be taken to advance our understanding of the non-reductive iron acquisition pathway in *P. tricornutum*: we highlight our work in the context of nascent knowledge about intracellular iron trafficking and chloroplast iron uptake in single-celled eukaryotic phytoplankton: we highlight the methodological advance and link it to fundamental outstanding questions in the field.

We feel strongly about keeping the last paragraph and the associated figure (Figure 8—figure supplement 1) in the manuscript. Listing a couple of outstanding problems in diatom cell biology where APEX2 and other related molecular tools will be useful strengthens the motivation for implementing APEX2 in the first place and connects the present(ed) advances with our future vision.

Reference to the use of TurbolD (also in the Discussion) begs the question of why this approach was not used in the first place.

At the time when this work was conceptualized and funding obtained (Spring 2015 and Fall 2015, respectively), only APEX, APEX2, and some pre-TurboID enzymes were available (we’d be happy to provide a detailed chronological list of all relevant studies). In addition, there were no reports on using these tools in phototrophic organisms.

TurboID only became available with Branon et al., 2018 which was followed by a few studies showing its utility in plants as cited in our manuscript. In short: TurboID did not exist. That said, as the diatom cell biology field moves forward, TurboID certainly has the potential to find its place alongside APEX2.

Again, I would recommend that the manuscript could be considerably focussed and shortened, focussing on the main key findings and reducing the overall levels of speculation.

We streamlined our manuscript and focused it on the key findings. Additional results and insights support our more focused treatment of the main text. Even though we draw inspiration from the transferrin pathway to help us contextualize our data, we distance ourselves from absolute parallels, and thereby leave it to the future studies to determine how related or unrelated the two pathways are.

Reviewer #2:I enjoyed reading the results of this study and am excited about this new chapter in understanding metal transport to complex plastids.1) The most exciting aspect of this study, surface-to-chloroplast iron trafficking, is also the least supported aspect of this study. The results are exciting but preliminary with regards to such a significant discovery. For instance, most of the hypothesized members in the "trafficking axis" have yet to be functionally characterized in algae and iron binding by the proposed chaperone, which has the most convincing chloroplast proximity, was not demonstrated. I suggest modifying the title of the manuscript to something like "Phytotransferrin endocytosis and the identification of a putative surface-to-chloroplast iron trafficking pathway".

We agree; we have been slightly too aggressive in choosing our initial title given the data at hand. The title has been toned down and is now not as definite as it was while still containing all the core points we want to highlight.

2) Unless I didn't find them in the supplementary material, only a single, 2D cell image is presented for each fusion construct. Multiple representative images should be provided. Also, because of the vacuole, the chloroplast, other organelles, and endosome-like vesicles could appear in very close proximity. A z-stack may help disambiguate the localization.

We now include a new supplementary figure (Figure 5—figure supplement 3) with three more representative images for each fusion construct. We realize further fluorescent microcopy experiments with other organellar markers and Z-stacks would be ideal, but this is either beyond the scope of our current work or was not possible given the Spring stay-at-home order and the lead author’s move away from San Diego, respectively.

Reference:

Liu X, Hempel F, Stork S, Bolte K, Moog D, Heimerl T, Maier UG, Zauner S. 2016. Addressing various compartments of the diatom model organism *Phaeodactylum tricornutum* via sub-cellular marker proteins. Algal Research 20:249–257.